# Long non-coding RNA produced by RNA polymerase V determines boundaries of heterochromatin

Gudrun Böhmdorfer[1], Shriya Sethuraman[2], M Jordan Rowley[1], Michal Krzyszton[3], M Hafiz Rothi[1], Lilia Bouzit[1], Andrzej T Wierzbicki[1]*

[1]Department of Molecular, Cellular, and Developmental Biology, University of Michigan, Ann Arbor, United States; [2]Bioinformatics Graduate Program, University of Michigan, Ann Arbor, United States; [3]Faculty of Biology, Institute of Genetics and Biotechnology, University of Warsaw, Warsaw, Poland

**Abstract** RNA-mediated transcriptional gene silencing is a conserved process where small RNAs target transposons and other sequences for repression by establishing chromatin modifications. A central element of this process are long non-coding RNAs (lncRNA), which in *Arabidopsis thaliana* are produced by a specialized RNA polymerase known as Pol V. Here we show that non-coding transcription by Pol V is controlled by preexisting chromatin modifications located within the transcribed regions. Most Pol V transcripts are associated with AGO4 but are not sliced by AGO4. Pol V-dependent DNA methylation is established on both strands of DNA and is tightly restricted to Pol V-transcribed regions. This indicates that chromatin modifications are established in close proximity to Pol V. Finally, Pol V transcription is preferentially enriched on edges of silenced transposable elements, where Pol V transcribes into TEs. We propose that Pol V may play an important role in the determination of heterochromatin boundaries.

*For correspondence: wierzbic@umich.edu

**Competing interests:** The authors declare that no competing interests exist.

## Introduction

RNA-mediated transcriptional gene silencing, in plants known as RNA-directed DNA methylation (RdDM), takes place in most eukaryotic organisms and results in heterochromatin formation by the deposition of DNA methylation and repressive histone modifications (*Holoch and Moazed, 2015*). This process relies on small RNAs, which usually are generated by the activities of an RNA-dependent RNA polymerase and Dicer. Small RNAs are subsequently incorporated into Argonaute and direct repressive chromatin modifications to complementary genomic regions (*Holoch and Moazed, 2015*). Recognition of target sequences by small RNAs requires ongoing non-coding transcription of the targets. This non-coding transcription gives rise to long non-coding RNA (lncRNA) which has been proposed to serve as a scaffold for Argonaute binding to chromatin, where incorporated small RNAs base pair with lncRNA (*Holoch and Moazed, 2015*).

In *Arabidopsis thaliana* this lncRNA is produced by a specialized DNA-dependent RNA polymerase, known as Pol V (*Haag and Pikaard, 2011*; *Wierzbicki et al., 2008*). Activity of Pol V is required for DNA methylation but not for the biosynthesis of the vast majority of small interfering RNAs (siRNAs) (*Kanno et al., 2005*; *Lahmy et al., 2009*; *Pontier et al., 2005*; *Wierzbicki et al., 2008*). This implicates lncRNAs produced by Pol V (referred to as Pol V transcripts) as a factor required for recognition of target loci by siRNAs. Pol V transcripts are believed to be capped or triphosphorylated on their 5' ends and not polyadenylated on their 3' ends (*Wierzbicki et al., 2008*). They associate with several RNA binding proteins, including ARGONAUTE 4 (AGO4) (*Wierzbicki et al., 2009*). It has been proposed that siRNAs incorporated into AGO4 base pair with Pol V transcripts and

recruit AGO4 to specific loci in the genome. Binding of AGO4 is followed by the binding of INVOLVED IN DE NOVO 2 (IDN2) which interacts with a subunit of the SWI/SNF ATP-dependent chromatin remodeling complex (*Böhmdorfer et al., 2014*; *Zhu et al., 2013*). Finally, Pol V transcripts, AGO4, and/or other associated factors recruit DOMAINS REARRANGED METHYLTRANSFE RASE 2 (DRM2), which is a *de novo* DNA methyltransferase (*Böhmdorfer et al., 2014*; *Gao et al., 2010*; *Zhong et al., 2014*). DNA methylation is then responsible for repression of Pol II transcription on silenced loci (*Lister et al., 2008*). While Pol V is involved in the late stages of the RdDM pathway, siRNA biogenesis starts with the activity of another specialized RNA polymerase, Pol IV (*Herr et al., 2005*; *Onodera et al., 2005*). Pol IV transcripts are substrates for RNA-DEPENDENT RNA POLY-MERASE 2 (RDR2) and DICER-LIKE 3 (DCL3), which produce 24nt siRNAs (*Haag et al., 2012*; *Li et al., 2015b*).

Despite being a central element of the RdDM pathway, Pol V transcripts are poorly understood. Unlike Pol IV transcripts, which are relatively abundant and have been characterized genome-wide (*Blevins et al., 2015*; *Li et al., 2015b*; *Zhai et al., 2015*), Pol V transcripts accumulate at low levels. This makes them difficult to detect using high-through sequencing approaches like RNA-seq. Therefore, it remains unknown if Pol V produces any RNAs beyond the very limited number of loci tested so far. It is also unclear what defines a Pol V promoter beyond published work suggesting that both Pol IV and Pol V are recruited by preexisting repressive chromatin modifications (*Johnson et al., 2014*; *Wierzbicki et al., 2012*; *Zhong et al., 2012*). It is further unknown which proteins interact with Pol V transcripts throughout the genome and how these proteins affect the transcripts. Additionally, the role of Pol V transcripts in forming the RdDM effector complex remains mysterious with several key mechanistic aspects being mostly based on speculation. These include the distance between the progressing polymerase and proteins binding to lncRNAs and the identity of nucleic acids base pairing with siRNAs (*Dalakouras and Wassenegger, 2013*; *Matzke et al., 2015*). Finally, it is unknown if and how the specificity of Pol V recruitment to chromatin targets RdDM to individual genomic regions.

## Results

### Genome-wide identification of transcripts associated with Pol V

Current knowledge of the *in vivo* functions of Pol V and RNAs produced by this polymerase is based on a very limited number of loci (*Wierzbicki et al., 2009*; *Zheng et al., 2013*; *Zhu et al., 2013*; *Böhmdorfer et al., 2014*; *Wierzbicki et al., 2008*). To overcome this limitation, we designed an experimental approach to identify Pol V transcripts throughout the *Arabidopsis* genome. We first enriched Pol V-associated RNAs using RNA immunoprecipitation with an antibody against NRPE1, the largest subunit of Pol V (*Pontier et al., 2005*; *Wierzbicki et al., 2009*), and then subjected the samples to high-throughput sequencing (Pol V RIP-seq). We performed these experiments in Col-0 wild-type and in the *nrpe1* mutant. This assay allowed the identification of genomic regions, where sequencing reads accumulated in Col-0 wild-type but not in *nrpe1* (*Figure 1A*, *Figure 1—figure supplement 1*). Hence, these reads originate from RNAs which are specifically associated with Pol V. Given that Pol V is a DNA-dependent RNA polymerase *in vitro* (*Haag et al., 2012*), these reads most likely stem from transcripts generated by Pol V. We also detected a considerable amount of signal over annotated genes, however, these remained unchanged in *nrpe1* (*Figure 1A*) and are therefore unlikely to be associated with Pol V. This signal was mostly present on active genes and indicates transcription by Pol I, II, III, and/or IV.

We used the RIP-seq data to annotate Pol V-associated RNAs genome-wide and identified 4502 individual high confidence Pol V-associated transcripts. Transcript calling used data from two independent biological replicates of RIP-seq. Data from both repeats were first combined to determine the ends of Pol V-associated RNAs. Then, read counts from both repeats were considered separately to filter the transcript list applying a combination of arbitrary criteria and statistical testing using the negative binomial test. The filtering criteria included a minimum of 8 reads in Col-0, a minimum four-fold enrichment between Col-0 and *nrpe1*, a p value of 0.05 and an FDR of 0.05. Details of the transcript calling strategy are described in the Materials and methods.

To visualize Pol V-associated RNAs, we plotted the average Pol V-RIP (Col-0/*nrpe1*) signal combined from both biological repeats on identified RNAs and their flanking regions (*Figure 1B*).

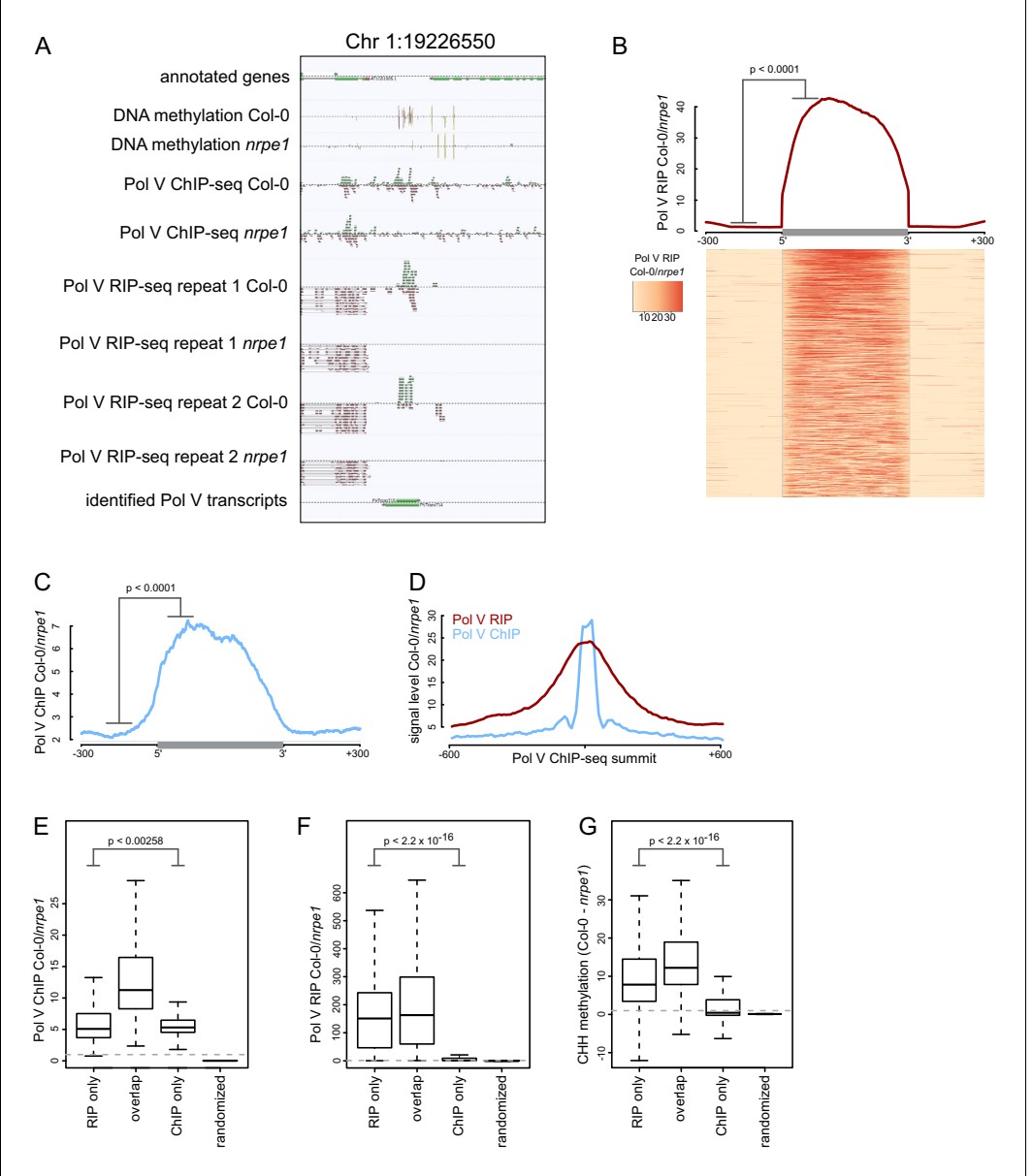

**Figure 1.** Genome-wide identification of RNA produced by Pol V (Pol V transcripts). (**A**) A genomic region giving rise to Pol V transcripts. The screenshot shows sequencing reads from both repeats of Pol V RIP-seq as well as Pol V ChIP-seq (*Wierzbicki et al., 2012*), DNA methylation (*Stroud et al., 2013*), and annotations of genes and Pol V transcripts. (**B**) Pol V RIP signal is largely limited to identified Pol V transcripts. All annotated Pol V transcripts were scaled to uniform lengths and average Pol V RIP signal from both biological repeats combined (Col-0/*nrpe1*, [RPM]) was plotted. The heatmap below shows Pol V RIP signal on individual transcripts sorted by length. The p value was calculated using the permutation test by comparing 100 nt long regions starting 200 nt upstream and 50 nt downstream of 5' ends of the annotated transcripts. (**C**) Pol V binding to chromatin is enriched on Pol V transcripts. Profile of average Pol V ChIP-seq signal (Col-0/*nrpe1* [RPM]) on scaled Pol V transcripts ± 300 bp. The p value was calculated using the permutation test by comparing 100 nt long regions starting 200 nt upstream and 50 nt downstream of 5' ends of the annotated transcripts. (**D**) Pol V RIP-seq signal is enriched on regions where Pol V binds chromatin. Profiles of average Pol V ChIP-seq signal (Col-0/*nrpe1*) and Pol V RIP signal (Col-0/*nrpe1*) on Pol V ChIP-seq peaks (*Wierzbicki et al., 2012*) aligned with their summits ± 600 bp (10 bp resolution). (**E–G**) Loci generating Pol V transcripts are bound by Pol V and are targets of RdDM. Boxplots show regions producing Pol V transcripts but not overlapping ChIP-seq peaks and *vice versa* (RIP only and ChIP only, respectively) and on Pol V transcript regions overlapping ChIP peaks (overlap). Significance has been tested using the Wilcoxon test. (**E**) Pol V ChIP-seq (Col-0/*nrpe1* [RPM]), (**F**) Pol V RIP-seq (Col-0/*nrpe1* [RPM]) and (**G**) CHH DNA methylation (Col-0 - *nrpe1*).
*Figure 1 continued on next page*

*Figure 1 continued*

The following figure supplement is available for figure 1:

**Figure supplement 1.** Genome-wide identification of RNA produced by Pol V (Pol V transcripts).

Individual transcripts were scaled to allow visualization over the entire lengths of the transcripts. We observed high levels of Pol V-associated transcription throughout the predicted transcripts in both replicates and only trace amounts of Pol V-associated transcription outside the annotations (*Figure 1B*). This was true for the vast majority of Pol V-associated transcripts as shown on the corresponding heatmap where every row represents an individual transcript (*Figure 1B*). This was also true when we analyzed both biological repeats separately (*Figure 1—figure supplement 1*). Reproducibility between biological repeats was further tested by comparing Pol V RIP-seq signal intensities on annotated Pol V-associated RNAs in both repeats (*Figure 1—figure supplement 1*). Signal strengths measured as differences between RPM normalized read counts in Col-0 and *nrpe1* were significantly correlated (Pearson correlation r = 0.719, $p < 2.2*10^{-16}$), which further increases confidence in the quality of our transcript calling. Taken together, we developed a strategy which allows the sensitive and reproducible identification of Pol V-associated RNAs throughout the genome.

## Pol V-associated RNAs are produced by Pol V

Transcripts identified using RIP-seq are bound by Pol V and are expected to be the products of Pol V based on its DNA-dependent RNA polymerase activity (*Haag et al., 2012*). However, they could also be produced by another RNA polymerase and bind to Pol V posttranscriptionally. To distinguish between these possibilities, we first checked if accumulation of those transcripts required Pol V. We tested several newly identified Pol V-associated transcripts using locus-specific RT-qPCR and identified 20 loci which were suitable for primer design and had a strong reduction of RNA accumulation in *nrpe1* in RT-qPCR (*Figure 1—figure supplement 1*). Therefore, transcripts obtained by RIP-seq are not only associated with Pol V but their accumulation also depends on Pol V, which indicates that these transcripts are products of Pol V.

Next, we tested if Pol V-associated transcripts are produced from regions where Pol V is bound to DNA. We compared our RIP-seq with previously published Pol V ChIP-seq obtained using the same antibody (*Wierzbicki et al., 2012*) and found that Pol V binds chromatin at regions where we detected Pol V-associated transcripts (*Figure 1C*). Pol V transcription was also enriched on many genomic regions bound by Pol V (*Wierzbicki et al., 2012*) (*Figure 1D*). Most regions producing Pol V-associated transcripts also displayed Pol V binding to chromatin and Pol V-dependent CHH methylation (*Stroud et al., 2013*) which is a hallmark of RdDM (*Figure 1E–G*). Regions identified only by ChIP-seq but not in RIP-seq had very low levels of Pol V-associated transcripts and low levels of Pol V-dependent CHH methylation (*Stroud et al., 2013*) (*Figure 1E–G*). In contrast, regions giving rise to Pol V-associated transcripts, which do not overlap Pol V ChIP-seq peaks, still displayed reduction of DNA methylation in *nrpe1* (*Figure 1G*), suggesting that RIP-seq is a much more sensitive approach for detecting genomic regions transcribed by Pol V. Overall, this analysis indicates that Pol V-associated transcripts are produced from regions bound by Pol V. Together with published *in vitro* data (*Haag et al., 2012*), these results suggest that Pol V-associated RNAs identified using RIP-seq are produced by Pol V transcribing a genomic DNA template. Therefore, these RNAs are likely to be *bona fide* Pol V transcripts.

## Pol V regulatory elements

RIP-seq identifies Pol V transcripts with a higher resolution than ChIP-seq and should facilitate discovery of the promoter of Pol V. RNA polymerases I, II, and III all use conserved sequence elements as their core promoters and, thus, Pol V may as well. Our attempts to identify conserved sequence motifs upstream of Pol V transcripts yielded no conclusive results. Although *de novo* discovery of promoter elements in plant genomes is not trivial (*Molina and Grotewold, 2005*), it is possible that Pol V may be directed to specific genomic loci by factors other than conserved sequence motifs.

To identify features that may guide Pol V, we first determined which categories of loci are transcribed by Pol V. Consistent with previously published data (*Zheng et al., 2013*; *Zhong et al., 2012*), Pol V transcripts originated from pericentromeric regions and from euchromatic chromosome arms (*Figure 2A*). Pol V transcripts were preferentially produced from intergenic regions, gene promoters, and all transposon families except long terminal repeat (LTR) transposons (*Figure 2B*). This distribution is consistent with previous reports suggesting that preexisting repressive chromatin modifications are necessary for Pol V activity (*Kuhlmann and Mette, 2012*; *Johnson et al., 2014*; *Liu et al., 2014*).

CG methylation is required for efficient Pol V binding to chromatin (*Johnson et al., 2014*). To test if Pol V transcription overlaps CG methylation, we analyzed published whole genome bisulfite sequencing datasets (*Stroud et al., 2013*). DNA methylation in the CG context was increased throughout the genomic regions transcribed by Pol V but not outside of those regions (*Figure 2D*). Even though most Pol V transcripts were enriched in MET1-dependent CG methylation, the levels of CG methylation were not sufficient to predict the levels of Pol V transcription (*Figure 2—figure supplement 1*). This observation shows that CG methylation and Pol V transcription overlap and provides support for preexisting CG methylation being an important factor in guiding Pol V to specific loci. However, it also indicates that CG methylation does not regulate the level of Pol V transcription and that CG methylation is not needed upstream of transcription initiation sites. Instead it may be required within the transcribed regions.

We also tested if Pol V transcription overlaps with various posttranslational histone modifications (*Moissiard et al., 2012*; *Greenberg et al., 2013*; *Luo et al., 2013*) and found that H3K9me2 overlapped Pol V-transcribed regions in a way similar to CG methylation, while H3K4me2 was depleted on Pol V-transcribed regions (*Figure 2E*). H3K4me3, H3K36me3, and H3K9ac also appeared depleted but due to a higher noise level this depletion was not significant (*Figure 2—figure supplement 1*). Although it is unknown which histone modifications are controlling Pol V transcription and which are established in a Pol V-dependent manner, this is consistent with Pol V being guided to its genomic targets by repressive chromatin modifications present within the transcribed regions.

The overlap between CG methylation and Pol V transcription suggests that CG methylation may be required for Pol V transcription. To test this possibility, we assayed the accumulation of six individual Pol V transcripts in the *met1* mutant and *suvh4/5/6* triple mutant (*Figure 2—figure supplement 1*). Five loci showed a significant reduction in the accumulation of Pol V transcripts in the *met1* mutant, which is consistent with a requirement of MET1 for Pol V transcription and with previous reports (*Johnson et al., 2014*). The *suvh4/5/6* mutant had a reduced accumulation of Pol V transcripts at two loci (*Figure 2—figure supplement 1*), which indicates that SUVH4, 5, and 6 (and H3K9me2 they presumably establish) may have a more subtle or locus-specific effect on Pol V transcription. One of the tested loci showed a significant increase in RNA accumulation in both mutants (*Figure 2—figure supplement 1*), which may be attributed to the loss of Pol II silencing. These results are consistent with CG methylation being required for Pol V transcription.

The requirement of CG methylation for Pol V binding to chromatin and transcription may be interpreted as evidence of CG methylation recruiting Pol V to specific loci in the genome. This would predict that CG methylation should be sufficient for Pol V transcription. Alternatively, CG methylation may be one of many factors working together to determine the specificity of Pol V. To distinguish between these possibilities, we analyzed protein-coding genes, which show gene body CG methylation (*Bewick et al., 2016*). Pol V transcripts were significantly depleted on body-methylated genes, identified by high levels of CG methylation and low levels of CHH methylation (*Figure 2C*). In contrast, Pol V transcripts were enriched on genes having high levels of CHH methylation (*Figure 2C*), which are likely caused by intronic transposons (*Saze et al., 2013*). This suggests that CG methylation is not sufficient for Pol V transcription and consequently, CG methylation is one of many factors involved in guiding Pol V to specific genomic loci.

The possibility that preexisting repressive chromatin modifications guide Pol V predicts that this polymerase should transcribe both strands of DNA. To test this prediction, we plotted forward and reverse Pol V-RIP signal on aligned and scaled Pol V transcripts. We found that, indeed, Pol V transcribed bidirectionally on annotated Pol V transcripts (*Figure 2F*) and that transcription levels on both strands were somewhat correlated (*Figure 2—figure supplement 1*). Furthermore, transcription on both strands was shifted with annotated Pol V transcripts on the reverse strand starting ~51 bp before annotated transcripts on the forward strand end (*Figure 2—figure supplement 1*). These

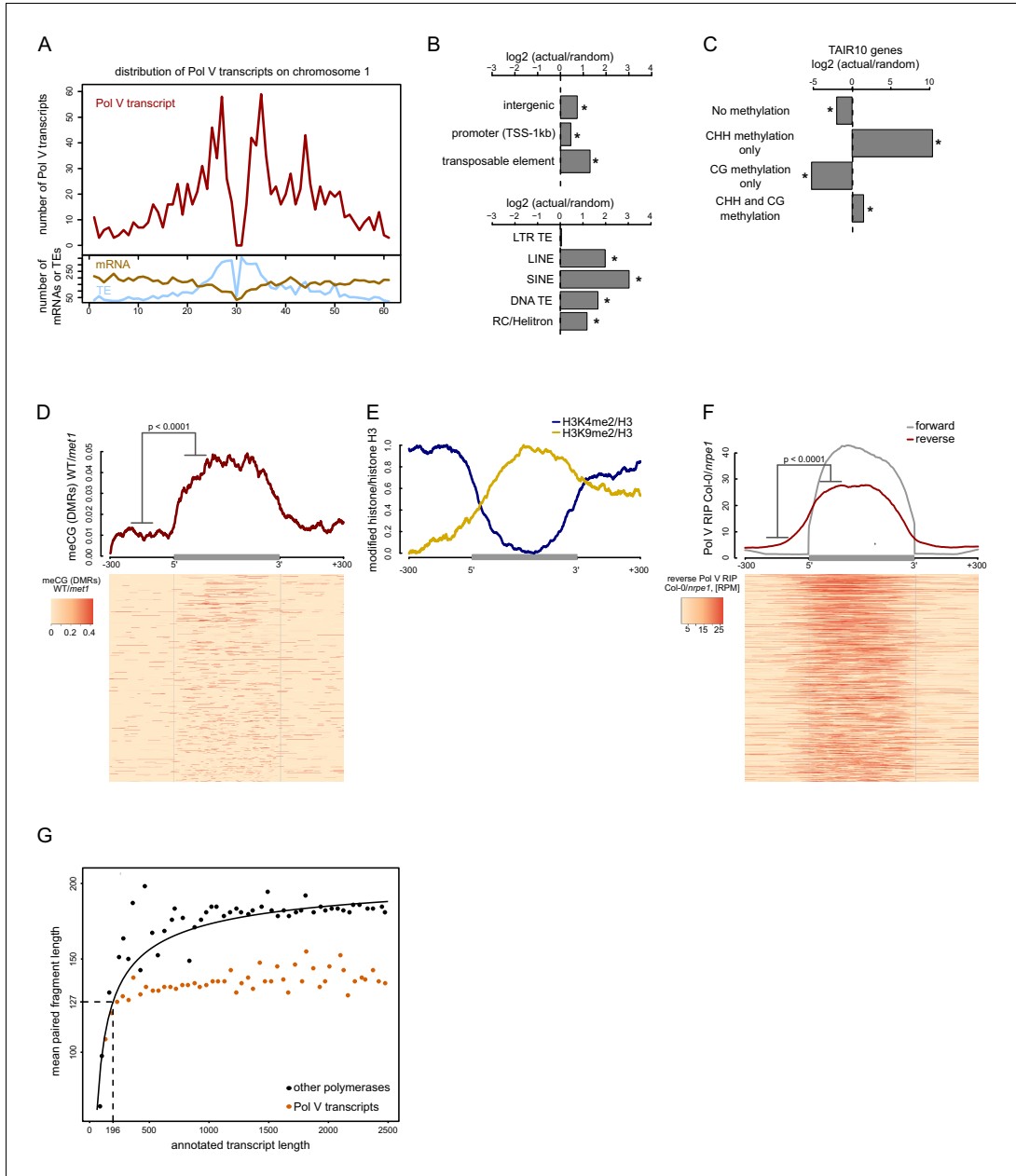

**Figure 2.** Pol V regulatory elements. (**A**) Pol V transcripts are produced from both pericentromeric regions and chromosome arms. The number of mRNAs, transposons (TAIR10) or Pol V transcripts was plotted on chromosome 1 in 500 kb windows. (**B**) Pol V transcripts are significantly enriched on promoters, intergenic sequences, and transposons of all families except LTR transposons. Plots show ratios of features overlapping Pol V transcripts to those overlapping randomized genomic regions. Promoters are defined as regions 1 kb upstream of the transcription start site of genes. Stars denote significant differences based on permutations (p<0.001). (**C**) CG methylation is not sufficient to mediate Pol V transcription. Genes annotated in TAIR10 were split into four categories based on the presence of CHH methylation (greater than 2%) and CG methylation (greater than 10%). Enrichment of annotated Pol V transcripts on those categories of genes was calculated by comparing the actual overlap with overlaps of random genomic loci. Stars denote p<0.004. (**D**) MET1-dependent CG methylation is enriched within Pol V-transcribed regions. Average CG methylation levels (*Stroud et al., 2013*) within differentially methylated regions (DMRs) were plotted on scaled Pol V transcripts. The p value was calculated using the permutation test by comparing 100 nt long regions starting 200 nt upstream and 50 nt downstream of 5' ends of the annotated transcripts. (**E**) A repressive histone modification is enriched on Pol V transcribed regions. Profiles of average enrichment of the modified histone (H3K9me2 and H3K4me2) over histone H3 were plotted on scaled Pol V transcripts ± 300 bp. Enrichment of H3K9me2 and depletion of H3K4me2 were statistically significant (p<0.0066

*Figure 2 continued on next page*

*Figure 2 continued*

and p<0.0001, respectively; permutation test). (**F**) Pol V transcribes bidirectionally. Profiles of averaged Pol V RIP-seq signal (Col-0/*nrpe1*) in forward (grey, *Figure 1B*) or reverse orientation (red) on scaled Pol V transcripts ± 300 bp. Forward strand refers to annotated transcripts, reverse strand refers to the strand opposite to the annotated transcripts. The p value was calculated using the permutation test by comparing 100 nt long regions starting 200 nt upstream and 50 nt downstream of 5' ends of the annotated transcripts. The shift between forward and reverse strands is further analyzed in *Figure 2—figure supplement 1*. (**G**) Annotated Pol V transcripts are composed of multiple shorter RNAs. Lengths of paired-end RNA fragments sequenced in RIP-seq mapping to nuclear and organellar genes (TAIR10) or to Pol V transcripts were compared to sizes of full length RNAs derived from annotations (TAIR10). Length of Pol V-bound RNAs (vertical dashed line) was estimated based on the median length of paired-end sequencing fragments mapping to Pol V transcripts (horizontal dashed line) and regression analysis of genes found in TAIR10 annotation (black line).

The following figure supplement is available for figure 2:

**Figure supplement 1.** Pol V regulatory elements.

results support the idea that internal chromatin modifications may be important for Pol V transcription. Furthermore, they also suggest that transcription on one strand and, possibly, subsequently deposited chromatin modifications may be important for the initiation of transcription on the other strand.

If internal repressive chromatin modifications control Pol V transcription, one might predict that Pol V could have multiple transcription initiation sites within one transcribed region. This would manifest itself in the presence of several shorter RNAs within most annotated Pol V transcripts. Alternatively, if Pol V had an external promoter, we would expect the presence of one predominant RNA with a transcription start site close to the beginning of the annotated transcript. Our RIP-seq protocol included sonication and random priming steps, which preclude us from directly capturing the ends of intact RNAs. However, we performed paired-end RNA sequencing, which provides an alternative way to distinguish between these two possibilities. To do so, we mapped the paired-end reads to all *Arabidopsis* transcripts annotated in TAIR10 as well as to Pol V transcripts. We then plotted the relationship between the transcript length and the mean size of paired-end sequenced RNA fragments mapping to this transcript (*Figure 2G*). As the RIP-seq datasets include significant amounts of background reads originating from polymerases other than Pol V, we were able to determine the relationship between the size of each transcript known from TAIR10 annotations and the mean length of mapped read-pairs (*Figure 2G*). Small Pol V transcripts also followed this relationship, however, longer Pol V transcripts did not produce longer sequenced fragments (*Figure 2G*). This indicates that actual RNAs produced from annotated Pol V transcripts are shorter than the size of these annotations. If the relationship between transcript length and sequenced RNA fragments is the same or at least similar for Pol V as it is for other DNA-dependent RNA polymerases, we would predict an RNA length of 196 nt based on the median paired end fragment obtained from the first repeat of Pol V RIP-seq (*Figure 2G*). A similar analysis of the second biological repeat of Pol V RIP-seq predicts a median RNA length of 205 nt. Considering that the median length of annotated Pol V transcripts is 689 nt (*Figure 2—figure supplement 1*), this indicates that annotated Pol V transcripts contain more than one transcription initiation and/or termination site. This is not only consistent with Pol V being controlled by internal promoters but also demonstrates that Pol V transcripts annotated in our study are not individual continuous transcriptional units but rather regions of Pol V transcriptional activity.

Overall, our analysis suggests that Pol V is controlled by internal promoters, similar to what has been reported for a subset of Pol III transcripts (*Geiduschek and Tocchini-Valentini, 1988*). Although any involvement of DNA sequence elements cannot be excluded at this time, our data are consistent with repressive chromatin modifications being at least important for Pol V recruitment and possibly working as a functional equivalent of a promoter.

## AGO4 binds most Pol V transcripts

Pol V is required for AGO4 binding to chromatin genome-wide (*Zheng et al., 2013*), however, it is unknown if AGO4 associates with Pol V transcripts beyond the handful of loci tested so far (*Wierzbicki et al., 2009*; *Böhmdorfer et al., 2014*). To test if the association with Pol V transcripts is a general feature of AGO4, we performed RIP-seq using an antibody against AGO4 (*Wierzbicki et al., 2009*) in Col-0 wild type, *ago4*, and *nrpe1*. Because the library prep method we used does not efficiently amplify siRNAs, this approach should specifically detect long RNAs associated with AGO4. AGO4 RIP-seq signal was significantly enriched on Pol V transcripts (*Figure 3A,B*) and this binding was dependent on Pol V (*Figures 3B*, *Figure 3—figure supplement 1*). The presence of AGO4 RIP-seq signal on most transcripts shown on the heatmap (*Figure 3A*) indicates that AGO4 associates with most if not all Pol V transcripts. This is further supported by a significant correlation between Pol V and AGO4 RIP-seq signals (*Figure 3—figure supplement 1*). Additionally, annotations based on AGO4 RIP-seq yielded transcripts with start and end sites similar to Pol V transcripts and overlapped regions with hallmarks of RdDM (*Lee et al., 2012*; *Stroud et al., 2013*; *Zheng et al., 2013*) (*Figure 3C*, *Figure 3—figure supplement 1*). It should be noted that the RIP assay includes formaldehyde crosslinking, which may preserve indirect interactions. Therefore, the association we observed may reflect direct physical interactions or indirect interactions with other proteins or nucleic acids in between AGO4 and lncRNAs.

AGO4 was shown to interact with Pol II on a limited number of loci where Pol II transcripts have been suggested to fulfill a role similar to Pol V transcripts (*Zheng et al., 2009*). To test if AGO4 binds to RNAs produced by polymerases other than Pol V, we identified RNAs whose association with AGO4 did not depend on Pol V. They were characterized by RIP signal present in Col-0 wild type and *nrpe1* but not in *ago4* (*Figure 3D*) and were depleted on transposons and enriched on intergenic sequences and promoters (*Figure 3E*). These RNAs were also enriched on Pol III-transcribed SINE elements (*Figure 3E*), which suggests that AGO4 may be binding Pol III transcripts. Regions generating those RNAs were not only not transcribed by Pol V but also failed to show any of the hallmarks of RdDM i.e. CHH methylation (*Stroud et al., 2013*), presence of siRNAs (*Lee et al., 2012*) or AGO4 binding to chromatin (*Zheng et al., 2013*) (*Figure 3D*). This suggests that association of AGO4 with RNAs produced by a polymerase other than Pol V does not lead to RdDM. Therefore, this interaction is either non-specific or reflects functions of AGO4 independent of RdDM. We conclude that AGO4 associates with most Pol V transcripts and may have an additional role not related to RdDM.

## AGO4 and IDN2 enhance the accumulation of Pol V transcripts

Widespread association of AGO4 with Pol V transcripts suggests that Pol V transcripts may be a substrate for AGO4 slicer activity (*Qi et al., 2006*). Alternatively, AGO4 could function without slicing Pol V transcripts. To distinguish between these possibilities, we performed RIP-seq with the anti-NRPE1 antibody in the *ago4* mutant. Presence of slicing by AGO4 would predict longer and more abundant transcripts in the *ago4* mutant. We observed none of those effects (*Figure 4A,B*, *Figure 4—figure supplement 1*). In contrast, accumulation of Pol V transcripts was decreased in the *ago4* mutant compared to Col-0 (*Figure 4A,B*, *Figure 4—figure supplement 1*). Moreover, analysis of the lengths of paired-end sequencing reads in the *ago4* mutant predicted an average transcript length of 200 nt, which is very similar to the size predicted for Col-0 wt (*Figure 2G*). These results indicate that AGO4 does not slice Pol V transcripts. Instead, AGO4 seems to enhance Pol V transcription or to stabilize Pol V transcripts. This is consistent with slicing activity being dispensable for Ago recruitment to chromatin in *S. pombe* (*Jain et al., 2016*). Alternatively, AGO4 slicing products originating from Pol V transcripts could be undetectable in our assay due to their size or loss of association with the Pol V complex.

Another protein shown to associate with Pol V transcripts is IDN2 (*Böhmdorfer et al., 2014*; *Zhu et al., 2013*). Although the biochemical function of IDN2 remains unknown, it could potentially affect the stability of Pol V transcripts. We tested this possibility by performing RIP-seq with the anti-NRPE1 antibody in the *idn2* mutant. Accumulation of Pol V transcripts was also reduced in *idn2* (*Figure 4A*, *Figure 4—figure supplement 1*) over the entire lengths of the annotated transcripts (*Figure 4B*). This indicates that, like AGO4, IDN2 also enhances Pol V transcription or increases the stability of Pol V transcripts on both strands (*Figure 4—figure supplement 1*). Effects observed in

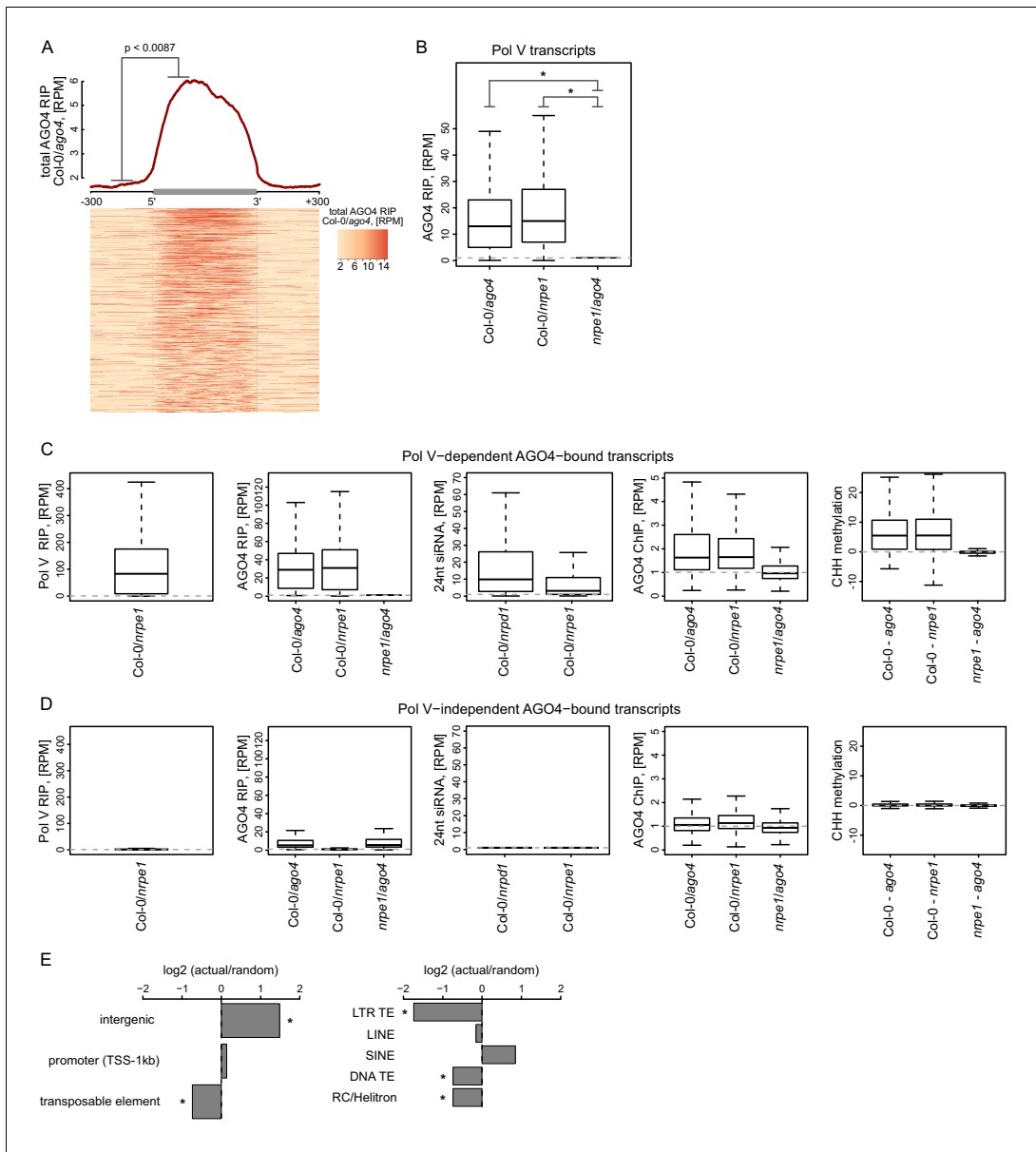

**Figure 3.** AGO4 binds most Pol V transcripts. (**A**) AGO4 RIP-seq signal (Col-0/*ago4*) is enriched on the majority of Pol V transcripts. Total AGO4 RIP signal was plotted on scaled Pol V transcripts. The p value was calculated using the permutation test by comparing 100 nt long regions starting 200 nt upstream and 50 nt downstream of 5' ends of the annotated transcripts. (**B**) Binding of AGO4 to Pol V transcripts depends on Pol V. The box plot shows AGO4 RIP-seq signal on Pol V transcripts. Stars denote $p<2.2 * 10^{-16}$ (Wilcoxon test). (**C**) Pol V-dependent association of AGO4 with RNA is correlated with RdDM. Boxplots show signal levels for Pol V RIP-seq, AGO4 RIP-seq, 24nt siRNA, AGO4 ChIP-seq, and CHH methylation. Transcript were called using AGO4 RIP-seq Col-0/*ago4* and considered Pol V-dependent if AGO4 RIP-seq Col-0/*nrpe1* $\geq$ 4. (**D**) Pol V-independent association of AGO4 with RNA is not correlated with RdDM. Box plots show signal levels for Pol V RIP-seq, AGO4 RIP-seq, 24nt siRNA, AGO4 ChIP-seq, and CHH methylation. Transcripts were called using AGO4 RIP-seq Col-0/*ago4* and considered Pol V-independent if AGO4 RIP-seq *nrpe1/ago4* $\geq$ 4. (**E**) Pol V-independent transcripts bound by AGO4 are enriched on intergenic sequences but are depleted on all transposons except SINEs. Plots show ratios of features overlapping transcripts to those overlapping randomized transcripts. Stars denote p<0.001 (permutation test).

The following figure supplement is available for figure 3:

**Figure supplement 1.** AGO4 binds most Pol V transcripts.

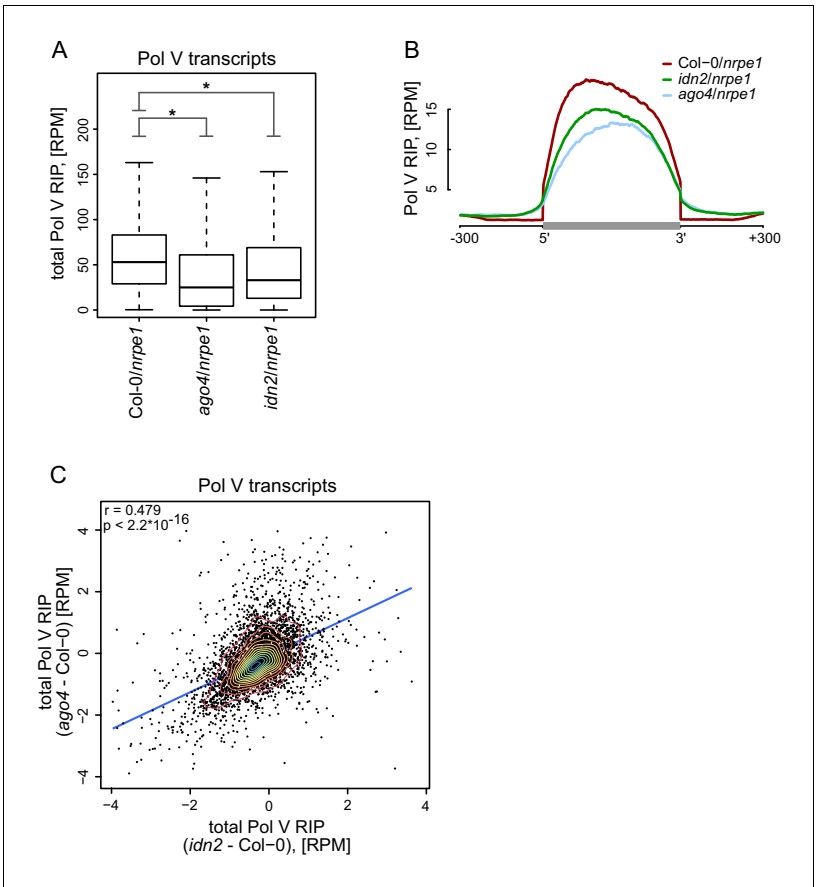

**Figure 4.** AGO4 and IDN2 enhance the accumulation of Pol V transcripts. (**A**) Accumulation of Pol V transcripts is reduced in *ago4* and *idn2*. Box plots show ratios of Pol V RIP-seq signals on Pol V transcripts in various genotypes. Stars denote p<2.2 * $10^{-16}$ (Wilcoxon test). (**B**) Accumulation of Pol V transcripts is reduced in *ago4* and *idn2* over the entire lengths of Pol V transcripts. Average Pol V RIP-seq enrichment was plotted on scaled Pol V transcripts. Differences between Col-0 and *ago4* as well as Col-0 and *idn2* are significant when measured between positions 50 nt and 150 nt downstream of 5' ends of Pol V transcripts (p<0.0001, permutation test). (**C**) On most Pol V transcripts, Pol V-transcription is affected in a similar way in *ago4* and *idn2*. Scatterplot of total Pol V RIP signal in *ago4* - Col-0 vs. *idn2* - Col-0. The plot shows a trend line calculated using linear regression (blue) as well as Pearson correlation coefficient and its p value.

The following figure supplement is available for figure 4:

**Figure supplement 1.** AGO4 and IDN2 enhance the accumulation of Pol V transcripts.

*ago4* and *idn2* were somewhat correlated (*Figure 4C*) and could not be explained by an overall reduction of RNA levels in *ago4* and *idn2* as we did not observe a similar decrease on mRNAs (*Figure 4—figure supplement 1*). This suggests that mutations in AGO4 and IDN2 could affect the stability of Pol V transcripts. Alternatively, these mutations could indirectly affect Pol V transcription by causing a general reduction in repressive chromatin modifications on Pol V transcribed regions, which, in turn, would reduce the rate of Pol V transcription. Overall, our results show that AGO4 and IDN2 are unlikely to contribute to slicing or other forms of degradation of Pol V transcripts.

## RdDM is restricted to Pol V-transcribed regions

The current models of RdDM show that AGO4, IDN2, and other RNA-binding proteins interact with Pol V transcripts at some distance from the transcribing core Pol V complex. This distance is allowed by the lengths of lncRNA and the C-terminal domain of Pol V which interacts with AGO4 (*Li et al., 2006*; *El-Shami et al., 2007*). Although this spatial separation between Pol V and downstream

factors has not been addressed experimentally, it predicts that the DNA methylation machinery may have some level of spatial flexibility especially over densely packed chromatin relative to the progressing position of the core Pol V polymerase complex. This flexibility could result in DNA methylation being established outside of the regions transcribed by Pol V. To test this possibility, we plotted Pol V-dependent CHH methylation (*Stroud et al., 2013*) on DNA sequences corresponding to Pol V transcripts. CHH methylation was significantly enriched within these sequences (*Figure 5A*, *Figure 5—figure supplement 1*). However, only trace levels of CHH methylation were observed outside (*Figure 5A*, *Figure 5—figure supplement 1*). This result shows that at least in genomic regions in close proximity to the annotated Pol V transcripts, the RdDM pathway is only able to deposit DNA methylation within regions transcribed by Pol V.

We further tested if other components of RdDM are present outside of DNA sequences corresponding to Pol V transcripts. Analysis of previously published small RNA datasets (*Lee et al., 2012*) indicated that Pol IV- and Pol V-dependent 24nt siRNAs mostly accumulated within regions transcribed by Pol V (*Figure 5B*, *Figure 5—figure supplement 1*), indicating that Pol IV and Pol V likely transcribe the same genomic regions. Similarly, AGO4 associated with chromatin (*Zheng et al.,*

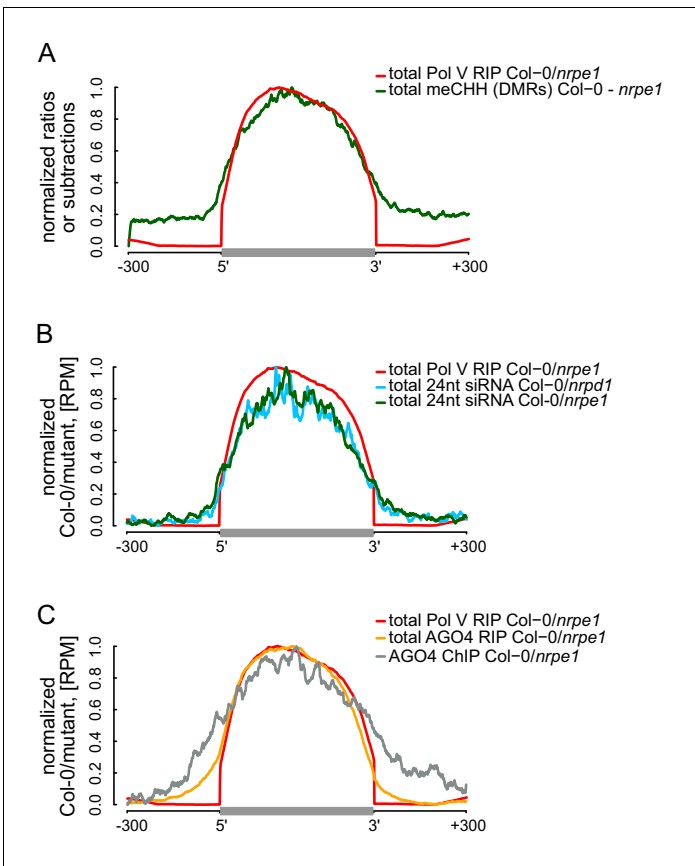

**Figure 5.** RdDM is restricted to Pol V-transcribed regions. (**A**) CHH methylation dependent on Pol V closely overlaps Pol V transcription. Average CHH methylation levels within differentially methylated regions (DMRs) were plotted on scaled Pol V transcripts ± 300 bp. Average Pol V RIP-seq signal (*Figure 1B*) was plotted as a reference. (**B**) siRNAs closely overlap Pol V transcription. Average enrichment of 24nt siRNA (Col-0/*nrpd1* and Col-0/*nrpe1*) was plotted on scaled Pol V transcripts. Average Pol V RIP-seq signal (*Figure 1B*) was plotted as a reference. (**C**) AGO4 binds to Pol V transcripts and corresponding DNA over the entire regions transcribed by Pol V. Average signals of AGO4 RIP-seq and AGO4 ChIP-seq Col-0/*nrpe1* were plotted on scaled Pol V transcripts. Average Pol V RIP-seq signal (*Figure 1B*) is shown as a reference.

The following figure supplement is available for figure 5:

**Figure supplement 1.** RdDM is restricted to Pol V-transcribed regions.

*2013*) mostly within Pol V transcribed regions (*Figure 5C*). These results show that, at least in genomic regions in close proximity to the annotated Pol V transcripts, several features of RdDM are predominantly restricted between the start and the end of Pol V transcription. Another and not mutually exclusive explanation of these results is that Pol V transcription is tightly limited to regions with preexisting DNA methylation. Both interpretations would be inconsistent with models assuming that the flexibility of Pol V transcripts could allow chromatin modifying enzymes to reach outside of the transcribed regions. Overall, these results are consistent with Pol V transcripts working in *cis* and mediating repressive chromatin modifications exclusively within transcribed regions.

## Strand bias of RdDM

The observed high spatial resolution of RdDM could be accompanied by a correlation between strand preference of Pol V transcripts and CHH methylation. Alternatively, Pol V transcripts may mediate the establishment of CHH methylation on both strands of DNA. Although Pol V tends to transcribe both strands of DNA (*Figure 2F*), these two scenarios can be distinguished because the levels of Pol V transcription are often not equal between both strands (*Figure 2—figure supplement 1*). To test if RdDM displays a strand preference on Pol V-transcribed regions, we separately plotted both strands of siRNAs (*Lee et al., 2012*) (*Figure 6A,B*) and DNA methylation (*Stroud et al., 2013*) (*Figure 6C*) on Pol V-transcribed sequences. We observed no differences in mean signal strengths on either strand throughout the transcribed regions. We also found no correlations in strand preference between Pol V transcription, CHH methylation (*Stroud et al., 2013*), and siRNAs (*Lee et al., 2012*) on Pol V transcribed regions (*Figure 6—figure supplement 1*). This indicates that there is no strand preference of DNA methylation relative to siRNAs or Pol V transcripts, which indicates that Pol V transcripts mediate the establishment of CHH methylation on both strands of DNA.

## Importance of AGO4 binding to Pol V transcripts

Several key RdDM factors have been shown to interact with both DNA and RNA on silenced loci (*Böhmdorfer et al., 2014*; *Wierzbicki et al., 2009*). It remains unknown, which interaction is more important for RdDM. To answer this question, we focused our analysis on heterochromatic domains where protein binding to DNA may be constrained by a high density of nucleosomes but Pol V can still transcribe. We identified heterochromatic domains with high density of H3K9me2 (*Moissiard et al., 2012*) and plotted various features of RdDM over their edges. Both Pol V binding to chromatin reported by ChIP-seq (*Wierzbicki et al., 2012*) and Pol V transcription observed by RIP-seq overlapped the heterochromatic domains, which is consistent with Pol V transcribing silenced genomic regions (*Figure 6D*). CHH methylation (*Stroud et al., 2013*) was also enriched on the heterochromatic domains (*Figure 6E*). Similarly, AGO4 binding to Pol V transcripts observed by RIP-seq overlapped H3K9me2 (*Moissiard et al., 2012*), Pol V transcription, 24nt siRNAs (*Lee et al., 2012*), and CHH methylation (*Stroud et al., 2013*) (*Figure 6D–G*). Interestingly AGO4 binding to DNA observed by ChIP-seq (*Zheng et al., 2013*) was strongly enriched on chromatin flanking these heterochromatic domains (*Figure 6G*). This indicates that CHH methylation more closely follows AGO4 binding to Pol V transcripts than to DNA, suggesting that AGO4 interaction with Pol V transcripts may be the primary event directing downstream factors of RdDM.

Binding of AGO4 to chromatin outside of heterochromatic domains could be explained by exclusion of protein binding to DNA by nucleosomes. To test this possibility, we grouped heterochromatic domains by strength of H3 ChIP-seq signal (*Moissiard et al., 2012*), which should indicate nucleosome density. Indeed, AGO4 binding (*Zheng et al., 2013*) outside of the heterochromatic regions was correlated with nucleosome density (*Figure 6H*, *Figure 6—figure supplement 1*).

Overall, these results show the central role of AGO4 interactions with Pol V transcripts in RdDM. Although the exact positions of various RdDM components within the silencing machinery and potential effects of formaldehyde crosslinking remain unknown, our data are consistent with a speculative model where downstream components of RdDM interact with Pol V transcripts in direct proximity to the Pol V complex and siRNAs base pair with RNA exiting the Pol V complex.

## Pol V determines the edges of transposons

RdDM targets edges of transposons while interior regions of large transposons are silenced by other epigenetic mechanisms (*Zemach et al., 2013*). However, current mechanistic understanding of

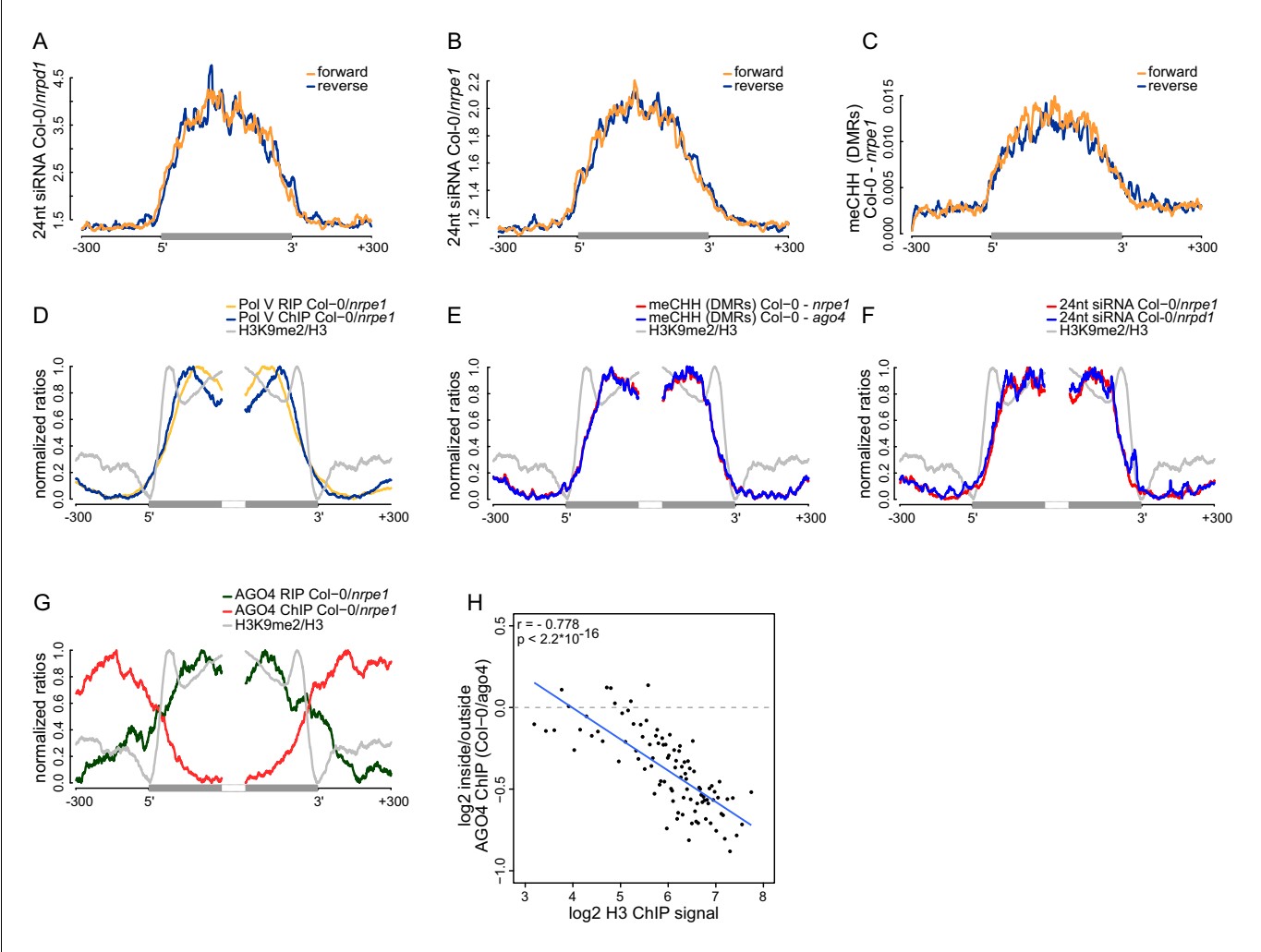

**Figure 6.** Strand bias of RdDM and importance of AGO4 binding to Pol V transcripts. (**A**) Pol IV-dependent 24nt siRNAs do not show a strand bias on Pol V transcripts. Average signal (Col-0/*nrpd1*, [RPM]) for reads with the same or the opposite orientation as Pol V transcripts was plotted on scaled Pol V transcripts ± 300 bp. (**B**) Pol V-dependent 24nt siRNAs do not show a strand bias on Pol V transcripts. Average signal (Col-0/*nrpe1*, [RPM]) for reads with the same or the opposite orientation as Pol V transcripts was plotted on scaled Pol V transcripts. (**C**) CHH methylation does not show a strand preference on Pol V transcripts. Average signal of called differentially methylated regions (DMRs) for CHH methylation (Col-0 - *nrpe1*) from the same or opposite strand as the Pol V transcript was plotted on scaled Pol V transcripts. (**D–G**) CHH methylation follows AGO4 interactions with Pol V transcripts on the edges of heterochromatic domains. (**D**) Pol V binds and transcribes DNA at the edges of heterochromatic domains. (**E**) CHH methylation is deposited on regions transcribed by Pol V. (**F**) 24nt siRNAs overlap Pol V transcripts. (**G**) AGO4 associates with RNA on regions transcribed by Pol V but association of AGO4 with DNA detectable by ChIP-seq is present outside of the heterochromatic domains. Profiles represent normalized average signals on heterochromatic domains (with H3K9me2) ± 300 bp, aligned at the ends. In each panel, gray bars on the x-axis (H3K9me2 region) and gray profiles (H3K9me2/H3) are shown. (**H**) High nucleosome density prevents AGO4 from binding to DNA within heterochromatic domains. Scatterplot compares H3 ChIP-seq signal to AGO4 ChIP-seq signal outside or inside of heterochromatic domains. Heterochromatic domains were combined in 100 groups based on their H3 levels and plotted against the log2 value of AGO4 ChIP-seq inside/outside the H3K9me2 region. 'Outside' was defined as the 50 to 250 bp upstream of the left end of the heterochromatic domain, while 'inside' corresponds to 50 to 250 bp inside the heterochromatic domain. The plot shows a trend line calculated using linear regression (blue) as well as Pearson correlation coefficient and its p value.

The following figure supplement is available for figure 6:

**Figure supplement 1.** Strand bias of RdDM and importance of AGO4 binding to Pol V transcripts.

RdDM does not explain this preference towards the edges of transposons. One possibility is that Pol V preferentially transcribes the edges of transposons. Alternatively, Pol V could transcribe the entire lengths of transposons but siRNAs could only be produced on the edges. To distinguish between these possibilities, we plotted Pol V RIP-seq data on all transposons which overlap annotated Pol V transcripts. While short transposons were entirely transcribed by Pol V, longer TEs had a strong enrichment of Pol V transcription on their edges (*Figure 7A*). We further analyzed euchromatic transposons longer than 4 kb, similar to those studied by (*Zemach et al., 2013*). They also displayed a strong enrichment of Pol V transcription and Pol V-dependent CHH methylation (*Stroud et al., 2013*) on their edges (*Figure 7B*, *Figure 7—figure supplement 1*). In contrast, regions inside the transposons appeared to be depleted in Pol V transcription compared to regions outside (*Figure 7B*). Pol IV transcripts (*Blevins et al., 2015*) were also enriched on edges of both categories of transposons, however, they were not depleted inside the transposons (*Figure 7—figure*

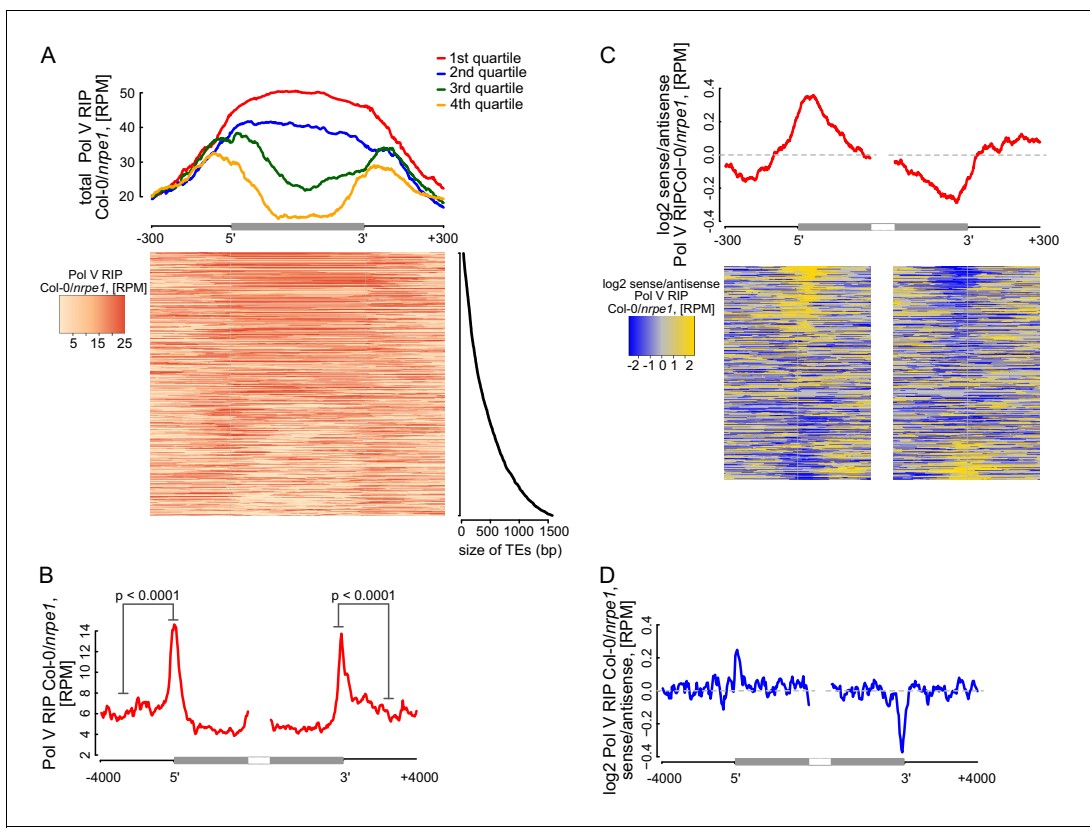

**Figure 7.** Pol V determines the edges of transposons. (**A**) Pol V transcripts are produced over the entire lengths of small transposons but are enriched at the edges of larger transposons. Annotated transposons (TAIR10) overlapping Pol V transcripts were split into quartiles according to their size (smallest to largest), scaled and the Pol V RIP-seq signal Col-0/*nrpe1* was plotted. Heatmap shows individual transposons sorted by size (sizes shown on the adjacent plot). (**B**) Pol V transcription is enriched on the edges of large transposons. Annotated euchromatic transposons greater than 4 kb were aligned by their 5'- and 3'-ends and average Pol V RIP-seq signal was plotted in 50 bp windows. The p value was calculated using the permutation test by comparing 500 nt long regions starting 1000 nt outside and 250 nt within the TEs. (**C**) Pol V transcribes into transposons. Transposons used in *Figure 7A* were aligned by their 5'- and 3'-ends and the average ratio of sense to antisense Pol V RIP-seq signal was plotted. Heatmaps show individual transposons sorted by the strength of transcription into the TEs. (**D**) Pol V transcribes into transposons. Transposons used in *Figure 7B* were aligned with their ends and the average ratio of sense to antisense Pol V RIP-seq signal was plotted.

The following figure supplement is available for figure 7:

**Figure supplement 1.** Pol V determines the edges of transposons.

*supplement 1*). These results indicate that transcription by both Pol IV and Pol V are involved in targeting RdDM to the edges of transposons.

Preferential transcription of transposon edges by Pol IV and Pol V suggests that these polymerases may be involved in determining the borders of silenced regions. Little is known about the mechanisms determining chromatin boundaries in plants, however, transcription is inherently directional and therefore could be involved in this process (*Cohen and Jia, 2014*). Although Pol V tends to transcribe both strands of DNA (*Figure 2F*), a subset of Pol V transcripts is enriched on one strand (*Figure 6—figure supplement 1*) indicating that limited strand preference of Pol V may be involved in determining boundaries of heterochromatin. To test this hypothesis, we determined the ratio of strand preference of Pol V transcription relative to the orientation of transposons. We first analyzed transposons selected for the presence of overlaps with Pol V transcripts. Pol V transcription on the 5'-ends of transposons was enriched on the sense strand while Pol V transcription on the 3'-ends of transposons was enriched on the antisene strand. This indicates that Pol V transcripts showed an enrichment of Pol V transcription into the transposons at both ends (*Figure 7C*). Similarly, euchromatic transposons longer than 4 kb also showed enrichment of Pol V transcription into the transposons at both ends (*Figure 7D*). In contrast, datasets of Pol IV transcripts (*Blevins et al., 2015*) did not show any evidence of strand preference (*Figure 7—figure supplement 1*). This is consistent with the tight physical and functional association of Pol IV with RDR2 (*Haag et al., 2012*; *Law et al., 2011*), which promptly converts Pol IV transcripts into double stranded RNA. Because accumulation of Pol IV transcripts requires RDR2 (*Blevins et al., 2015*), the existence of strand preference of Pol IV transcription remains unknown.

These results indicate that Pol V preferentially transcribes into transposons. Therefore, we propose that Pol V may play a key role in determining the boundaries of heterochromatin by transcribing into silenced regions.

## Discussion

### Regulation of Pol V transcription

Pol V is one of the critical factors determining the specificity of RdDM. Therefore, the promoter of Pol V and mechanisms regulating Pol V transcription may determine which regions of the genome are targeted by RdDM. Previous studies using ChIP-seq failed to identify conserved sequence elements which could be the Pol V promoter (*Wierzbicki et al., 2012*; *Zhong et al., 2012*). Despite much higher resolution of RIP-seq we also did not identify any conserved external or internal sequence elements. Although *de novo* discovery of promoter elements is difficult even on Pol II genes (*Molina and Grotewold, 2005*) and requirement of short or variable sequence elements for Pol V transcription cannot be excluded, this is consistent with recent work showing the requirement of methyl-binding proteins SUVH2 and SUVH9 for Pol V binding to chromatin and transcription (*Johnson et al., 2014*; *Liu et al., 2014*). Since SUVH2 and SUVH9 bind methylated DNA through their SRA domains (*Johnson et al., 2008*, *2014*), this indicates that pre-existing DNA methylation may serve as an equivalent of the Pol V promoter. We provide additional support for that possibility by showing a strong overlap between CG methylation and Pol V transcription. At the same time the accumulation of Pol V transcripts is not correlated with the levels of CG methylation and CG methylation is not sufficient to recruit Pol V. Moreover, Pol V transcription is limited to the edges of transposons, while DNA methylation usually spans the entire lengths of the transposable elements. This indicates that other unknown factors are involved in the determination and regulation of Pol V transcription.

RNA polymerases I and II primarily use promoters located upstream of the transcription start sites, while Pol III mostly but not exclusively uses internal promoters (*Geiduschek and Tocchini-Valentini, 1988*). Pol V seems to behave more like Pol III in being controlled by features present within the transcribed regions. This means that, although Pol V is derived from Pol II (*Tucker et al., 2010*), it possibly uses a different transcription initiation machinery.

### The RdDM effector complex

Our data also provide insights into the interplay between Pol V, Pol V transcripts, and associated proteins, which we refer to as the RdDM effector complex. First, our data demonstrate on a

genome-wide scale that RdDM of endogenous loci is mostly restricted to regions transcribed by Pol V. This is consistent with data obtained in transgene- or virus-induced silencing (*Daxinger et al., 2009*; *Pélissier and Wassenegger, 2000*; *Sasaki et al., 2014*) and could be interpreted as evidence that siRNAs base pair with DNA and that Pol V only facilitates this interaction. Although this possibility cannot be excluded based on currently available data (*Matzke et al., 2015*), we provide additional indirect support of base pairing between siRNAs and nascent Pol V transcripts.

One reason why siRNAs-lncRNA base pairing seems to be a more likely scenario is lack of detectable strand preference between siRNAs, Pol V transcripts, and DNA methylation on Pol V-transcribed loci. Although such a preference has been reported on other categories of loci (*Lister et al., 2008*; *Zhong et al., 2014*), this discrepancy may be explained by the potential involvement of other RNA polymerases or other factors. Since we show that both strands of DNA are equally likely to be methylated even if siRNAs or Pol V transcripts show a strong strand preference, this could indicate that double stranded DNA is a substrate for DRM2. The simplest explanation for this observation would be base pairing between siRNAs and lncRNAs. Another result favoring siRNAs-RNA base pairing is our analysis of regions with high density of nucleosomes with H3K9me2. Although this analysis does not specifically take into account the activities of at least two putative ATP-dependent chromatin remodeling complexes involved in RdDM (*Kanno et al., 2004*; *Zhu et al., 2013*), it indicates that CHH methylation more closely follows AGO4 interactions with RNA than with DNA.

The exact architecture of the RdDM effector complex remains mysterious mostly because formaldehyde crosslinking does not allow distinguishing direct from indirect interactions. Our data show that this effector complex includes AGO4 throughout the genome, however AGO4 does not slice Pol V transcripts. Previous work indicates the involvement of IDN2 (*Böhmdorfer et al., 2014*; *Zhu et al., 2013*), which also seems to be involved in enhancing the accumulation of Pol V transcripts. If, as argued above, siRNAs incorporated into AGO4 base pair with Pol V transcripts, restriction of DNA methylation to Pol V-transcribed regions suggests that this base pairing is likely to occur in close physical proximity to the Pol V complex. This is consistent with observations that AGO4 physically interacts with the C-terminal domain of Pol V (*El-Shami et al., 2007*).

Another important question about the RdDM effector complex is the involvement of RNA polymerases other than Pol V, which has been suggested by genetic evidence (*Zheng et al., 2009*). Our data do not provide evidence of Pol I, II or III functionally substituting for Pol IV in the *nrpe1* mutant. However, it remains possible that these other polymerases may be involved in the initiation of transcriptional silencing or work with Argonaute proteins other than AGO4 (*Fultz et al., 2015*).

## Determination of heterochromatin boundaries

Although RdDM has been implicated in various biological processes (*Matzke and Mosher, 2014*), its functions remain to some extent mysterious. It is especially true for its presumably main role of silencing transposons. Transposons are stably silenced by the combined action of three silencing pathways. Two of them are maintenance pathways which rely on CG methylation maintained by MET1 (*Lister et al., 2008*) and non-CG methylation maintained by CMT2 and CMT3 (*Stroud et al., 2014*; *Zemach et al., 2013*). RdDM on the other hand is capable of establishing DNA methylation *de novo*. Recently, yet another pathway has been implicated in directing silencing to active transposons (*McCue et al., 2015*). Our data show that Pol IV and Pol V direct RdDM to edges of transposons. Moreover, Pol V preferentially transcribes into transposons, which indicates that Pol V may be involved in determining the boundaries of heterochromatin on transposable elements. This is reminiscent of the BORDERLINE lncRNA in *S. pombe* (*Keller et al., 2013*), however, unlike BORDERLINE, Pol V transcripts are more likely to maintain heterochromatin over the edges of transposons rather than to prevent the spreading of heterochromatin.

The role of RdDM in the determination of heterochromatin boundaries has been recently studied in maize, where mutations in RdDM components MOP1, MOP2, and MOP3 enhance spreading of euchromatin from genes into nearby transposons (*Li et al., 2015a*). Although this supports a role for RdDM in heterochromatin boundaries, the causal role of Pol V transcription in this process remains to be directly demonstrated.

The mechanism responsible for preferential Pol V transcription into transposable elements remains unknown. One possibility is that the transition from euchromatin to heterochromatin facilitates Pol V transcription. This is however inconsistent with our data showing that Pol V transcription is controlled by multiple internal promoter-like features. Another explanation is the presence of

more than one Pol V transcription initiation mechanism, with one mechanism responsible for unidirectional transcription on TE boundaries and the other mechanism mediating bidirectional transcription on all Pol V-transcribed loci.

## Materials and methods

### Plant material

Col-0 wild type, *nrpe1* (*nrpd1b-11*, [*Pontes et al., 2006*]), *ago4-1* (introgressed into the Col-0 background, [*Wierzbicki et al., 2009*; *Zilberman, 2003*]), *idn2-1* (*Ausin et al., 2009*), *met1-3* (*Saze et al., 2003*), and *suvh4R203/suvh5-2/suvh6-1* (*Ebbs and Bender, 2006*; *Pontvianne et al., 2012*) plants were grown in soil in long-day conditions. For all experiments, approximately 2.5 weeks old seedlings were used.

### Antibodies

The antibodies against the largest subunit of Pol V (NRPE1) or against AGO4 were described previously (*Pontier, 2005*; *Wierzbicki et al., 2009*).

### RT-qPCR and RIP-seq

RT-qPCR experiments were performed in biological triplicates as reported previously (*Rowley et al., 2013*). Oligonucleotides used for PCR are provided in *Supplementary file 1*. Fixation (0.5% formaldehyde) and RIP were performed according to the previously published protocol (*Rowley et al., 2013*) up to step 37, using optimized amounts of protein A agarose beads coated with salmon sperm DNA in case of RIPs performed with the α-NRPE1 antibody (*Pontier et al., 2005*) or Dynabeads protein A in case of α-AGO4 (*Wierzbicki et al., 2009*), respectively.

Next, 1/10th vol. 3M NaOAc (pH 5.3), 2.5 vol. 96% ethanol and 1 µl NF-Pellet Paint were added to the samples and the inputs, precipitated overnight at −80°C, and washed as described in steps 40 to 43 of the published protocol (*Rowley et al., 2013*). After resuspension in 10 µl miliQ water, the samples were digested with 5.8 u Turbo DNase I in the presence of 24 u of Ribolock RNase inhibitor at 25°C for 30 min. The reaction was stopped by adding 3 µl 25 mM EDTA (pH 8.0) and incubating at 65°C for 10 min. To ensure that the DNase I digest and the immuno-precipitation had been successful, 1/10th of the reaction was tested by RT-qPCR for the presence Pol V-transcripts and absence of genomic DNA as described in *Rowley et al. (2013)*. Minus-RT controls were included in all RT-qPCR assays and mock controls were included during protocol optimization experiments. The remaining 9/10th were precipitated and resuspended in 5 µl miliQ water for library generation.

Finally, library preparation was performed for mutant and wild-type samples by the University of Michigan Sequencing Core using the Illumina TruSeq Stranded mRNA Sample Prep Kit, replacing the heat fragmentation with an incubation step on ice for 5 min. No mRNA or rRNA depletion steps were performed. Libraries were sequenced by 50 bp paired-end sequencing.

### Mapping

Reads were mapped to the *Arabidopsis* genome (TAIR10) using SOAPsplice 1.10 (*Huang et al., 2011*), allowing a maximum gap size within a two segment alignment of 10300 bp (corresponding to the longest intron in *Arabidopsis*), choosing as output format SAM, and otherwise using default conditions. These conditions correspond to a maximum of 3 mismatches and 2 indels allowed and non-unique reads being mapped only once. Separately mapped reads belonging to the same pair were joined after mapping and only kept if reads from the same pair had mapped to the same chromosome and were at most 3 kb apart.

### RIP-seq transcript calling

We combined unique, non-genic (outside of TAIR10-annotated genes) sequencing reads from both biological repeats of RIP-seq. Regions with more than 8 reads positioned no further than 200 bp apart were identified as potential transcripts. We filtered the transcripts to keep only those with at least 1 read per 100 bp and with more than four fold enrichment (Col-0/*nrpe1*). The four fold enrichment test was then repeated with all sequencing reads, including not unique reads. Transcripts

containing genes annotated in TAIR10 have been removed. We then performed the negative bino-
mial test using NBPSeq R package (*Di et al., 2011*). Only transcripts with p<0.05 and FDR <0.05
were kept. We further filtered transcripts to keep only those with more than four fold enrichment
(Col-0/*nrpe1*) in each repeat counted separately and with more than 2 reads in Col-0 in each of the
biological repeats.

We called AGO4-bound transcripts as described for Pol V-transcripts, using the data obtained in
the AGO4 RIP-seq for Col-0 and *ago4*, except that we filtered to 4 reads in Col-0, six-fold enrich-
ment in Col-0/*ago4* and did not perform the negative binomial test (since we performed one biolog-
ical repeat of AGO4 RIP-seq). Transcripts were considered Pol V-dependent if the enrichment in the
AGO4-RIP (Col-0/*nrpe1*, [RPM]) was at least four-fold. AGO4-bound transcripts were considered to
be Pol V-independent if the AGO4-RIP *nrpe1/ago4* signal was at least four-fold.

## Heterochromatic regions

The genome was divided into 100 bp windows with 50 bp overlaps and H3K9me2 ChIP-seq
(*Moissiard et al., 2012*) reads were counted. Those windows containing a greater number of reads
than the median were kept and combined if sequentially located. Regions greater than 1 kb were
then used in the analysis. Pol V and AGO4 ChIP-seq (*Wierzbicki et al., 2012*; *Zheng et al., 2012*),
RIP-seq, siRNA (*Lee et al., 2012*), and CHH methylation (*Stroud et al., 2013*) data were plotted as
averages on aligned ends of heterochromatic regions ± 300 bp.

For comparing AGO4-ChIP (*Zheng et al., 2012*) and H3-ChIP (*Moissiard et al., 2012*) intensity
(*Figure 6H*, *Figure 6—figure supplement 1*), heterochromatic regions were grouped into 100 or 4
groups according to the strength of the H3 signal (left end + 50 to 250 bp). The log2 value of the
ratio of AGO4-ChIP signal inside (first 50 to 250 bp from the left end of the heterochromatic region)
and outside (region 250 to 50 bp upstream of the left end of the heterochromatic region) was calcu-
lated and plotted against the median H3-ChIP signal. To visualize the ranking and the binding of
AGO4 next to the H3K9me2 regions, the ranked heterochromatic regions were split into quartiles
and the average H3 ChIP-seq signal (*Moissiard et al., 2012*) or AGO4 ChIP-seq Col-0/*ago4* enrich-
ment (*Zheng et al., 2012*) was plotted as profiles at the 5'-end ± 300 bp of the called heterochro-
matic regions.

## Data visualization on heatmaps and profiles

Reads were counted using BEDTools 2.15.0 on Pol V-transcripts ± 300 bp and RPM normalized.
Transcripts were scaled to a uniform length and ratios of wild type and mutant signal was plotted on
scaled individual transcripts (heatmap) or as an average (profile). To allow the visualization of individ-
ual transcripts, heatmaps show every other transcript sorted by size. For DNA methylation
(*Stroud et al., 2013*), subtraction of mutant methylation levels from wild-type were used instead of
ratios and average methylation levels on differentially methylated regions (DMRs) were plotted.

Transposons annotated in TAIR10 were filtered for the presence of any overlaps with annotated
Pol V transcripts. Alternatively, transposons larger than 4 kb from genomic regions with more genes
than transposons were used. For strand preference analysis, ratios of the Pol V RIP-seq signal with
the same or the opposite orientation than transposons were plotted on scaled transposon overlap-
ping Pol V transcripts (± 300 bp) or, for transposons larger than 4 kb, at the 5'- and 3'-ends ± 4 kb.

Significance of differences observed on profiles of average signal strengths was tested using the
permutation test with 10,000 permutations. Averages from all nucleotides for specified regions were
calculated for each transcript/TE without scaling to uniform transcript lengths.

## Comparison of ends of Pol V transcripts

Differences between the positions of 5'- and 3'-ends of Pol V transcripts produced at the same locus
but from opposite strands were plotted as boxplots. For comparing Pol V transcripts and AGO4-
bound transcripts, the distances between ends were calculated for transcripts with the same orienta-
tion and a minimum overlap of 50%.

## Comparison of Pol V RIP-seq and Pol V ChIP-seq

Overlapping Pol V-transcripts were combined to Pol V-transcribed regions and compared to a previ-
ously published list of Pol V ChIP-seq peaks (*Wierzbicki et al., 2012*) obtained using the same anti-

NRPE1 antibody. Boxplots show enrichment on ChIP peaks or Pol V-transcripts or regions where both overlap. For randomization, 1000 permutations of the overlap of Pol V transcribed regions and Pol V ChIP peaks were performed.

### Prediction of transcript size

Pol V RIP-seq paired end sequencing reads were mapped to all *Arabidopsis* transcripts (TAIR10) and Pol V transcripts. Mean lengths of sequenced fragments were calculated for transcripts with distinct origins and annotated lengths. Regression analysis using transcripts annotated in TAIR10 was applied to predict the length of Pol V transcripts based on the median length of sequenced fragments.

### Previously published sequencing datasets

*Arabidopsis* genome annotations (TAIR10) were obtained from TAIR (www.arabidopsis.org). Pol V ChIP-seq data (SRA054962) and peak list, as well as, the AGO4 ChIP-seq data (GSE35381) were published previously (*Wierzbicki et al., 2012*; *Zheng et al., 2013*). DNA methylation data (GSE39901) were used from *Stroud et al. (2013)*. ChIP-seq data for histone modifications (GSE37644, GSE49090, and GSE28398) were published previously (*Greenberg et al., 2013*; *Moissiard et al., 2012*; *Luo et al., 2013*). Pol V ChIP-seq dataset in the *met1* mutant (GSE52041) was reported by (*Johnson et al., 2014*). siRNA (GSE36424) were reported previously (*Lee et al., 2012*). Pol IV transcription data (SRP059814) were used from (*Blevins et al., 2015*).

### Data access

The sequencing data from this study have been submitted to the NCBI Gene Expression Omnibus (GEO; http://www.ncbi.nlm.nih.gov/geo/) under accession number GSE70290.

## Acknowledgements

This work was supported by grants from the National Science Foundation (MCB 1120271) and the National Institutes of Health (R01GM108722) to ATW. GB was supported by the Austrian Science Fund (FWF) fellowship J3199-B09. MK was supported by the National Science Center, Poland (NCN) grant UMO-2013/08/T/NZ1/00313. MJR was partially supported by the University of Michigan Genetics Training Program (T32-GM07544). The content is solely the responsibility of the authors and does not necessarily represent the official views of the funding agencies.

## Additional information

### Funding

| Funder | Grant reference number | Author |
| --- | --- | --- |
| Austrian Science Fund | J3199-B09 | Gudrun Böhmdorfer |
| National Institute of General Medical Sciences | T32GM07544 | M Jordan Rowley |
| Narodowe Centrum Nauki | UMO-2013/08/T/NZ1/00313 | Michal Krzyszton |
| National Science Foundation | MCB 1120271 | Andrzej T Wierzbicki |
| National Institute of General Medical Sciences | R01GM108722 | Andrzej T Wierzbicki |

The funders had no role in study design, data collection and interpretation, or the decision to submit the work for publication.

### Author contributions

GB, Conception and design, Acquisition of data, Analysis and interpretation of data, Drafting or revising the article; SS, MJR, Analysis and interpretation of data, Drafting or revising the article; MK, MHR, LB, Acquisition of data; ATW, Conception and design, Analysis and interpretation of data, Drafting or revising the article

**Author ORCIDs**

Shriya Sethuraman, http://orcid.org/0000-0001-5033-1105

Andrzej T Wierzbicki, http://orcid.org/0000-0002-5713-1306

## Additional files

### Supplementary files

• Supplementary file 1. Oligonucleotides used in this study. The table shows oligonucleotides used for locus-specific qPCR-based assays used in this study.

### Major datasets

The following dataset was generated:

| Author(s) | Year | Dataset title | Dataset URL | Database, license, and accessibility information |
|---|---|---|---|---|
| Böhmdorfer G, Sethuraman S, Rowley MJ, Krzyszton M, Rothi MH, Bouzit L, Wierzbicki AT | 2016 | Long noncoding RNA produced by RNA Polymerase V determines boundaries of heterochromatin | http://www.ncbi.nlm.nih.gov/geo/query/acc.cgi?acc=GSE70290 | Publicly available at NCBI Gene Expression Omnibus (accession no: GSE70290) |

The following previously published datasets were used:

| Author(s) | Year | Dataset title | Dataset URL | Database, license, and accessibility information |
|---|---|---|---|---|
| Wierzbicki AT, Cocklin R, Mayampurath A, Lister R, Rowley MJ, Gregory BD, Ecker JR, Tang H, Pikaard CS | 2012 | Spatial and functional relationships among Pol V-associated loci, Pol IV-dependent siRNAs, and cytosine methylation in the Arabidopsis epigenome. | http://www.ncbi.nlm.nih.gov/sra/?term=SRA054962 | Publicly available at NCBI Sequence Read Archive (accession no: SRA054962) |
| Zheng Q, Rowley MJ, Bohmdorfer G, Sandhu D, Gregory BD, Wierzbicki AT | 2012 | RNA polymerase V targets transcriptional silencing components to promoters of protein-coding genes | http://www.ncbi.nlm.nih.gov/geo/query/acc.cgi?acc=GSE35381 | Publicly available at NCBI Gene Expression Omnibus (accession no: GSE35381) |
| Stroud H, Greenberg MVC, Feng S, Bernatavichute YV, Jacobsen SE | 2013 | Comprehensive analysis of silencing mutants reveals complex regulation of the Arabidopsis methylome. | http://www.ncbi.nlm.nih.gov/geo/query/acc.cgi?acc=GSE39901 | Publicly available at NCBI Gene Expression Omnibus (accession no: GSE39901) |
| Moissiard G, Cokus SJ, Cary J, Feng S, Billi AC, Stroud H, Husmann D, Zhan Y, Lajoie BR, McCord RP, Hale CJ, Feng W, Michaels SD, Frand AR, Pellegrini M, Dekker J, Kim JK, Jacobsen S | 2012 | MORC family ATPases required for heterochromatin condensation and gene silencing | http://www.ncbi.nlm.nih.gov/geo/query/acc.cgi?acc=GSE37644 | Publicly available at NCBI Gene Expression Omnibus (accession no: GSE37644) |
| Greenberg MVC, Deleris A, Hale CJ, Liu A, Feng S, Jacobsen SE | 2013 | Interplay Between Active Chromatin Marks and RNA-directed DNA Methylation in Arabidopsis thaliana | http://www.ncbi.nlm.nih.gov/geo/query/acc.cgi?acc=GSE49090 | Publicly available at NCBI Gene Expression Omnibus (accession no: GSE49090) |

| Luo C, Sidote DJ, Kerstetter RA, Michael TP, Lam E | 2013 | Histone modifications of Arabidopsis thaliana (aerial tissue) | http://www.ncbi.nlm.nih.gov/geo/query/acc.cgi?acc=GSE28398 | Publicly available at NCBI Gene Expression Omnibus (accession no: GSE28398) |
|---|---|---|---|---|
| Johnson LM, Du J, Hale CJ, Bischof S, Feng S, Chodavarapu RK, Zhong X, Marson G, Pellegrini M, Segal DJ, Patel DJ, Jacobsen SE | 2014 | SRA/SET domain-containing proteins link RNA polymerase V occupancy to DNA methylation | http://www.ncbi.nlm.nih.gov/geo/query/acc.cgi?acc=GSE52041 | Publicly available at NCBI Gene Expression Omnibus (accession no: GSE52041) |
| Lee T, Gurazada SGR, Zhai J, Li S, Simon SA, Matzke MA, Chen X, Meyers BC | 2012 | Characterization of small RNA-generating regions targeted by RNA polymerase V in Arabidopsis | http://www.ncbi.nlm.nih.gov/geo/query/acc.cgi?acc=GSE36424 | Publicly available at NCBI Gene Expression Omnibus (accession no: GSE36424) |
| Blevins T, Podicheti R, Mishra V, Marasco M, Wang J, Rusch D, Tang H, Pikaard CS | 2015 | Identification of Pol IV and RDR2-dependent precursors of 24 nt siRNAs guiding de novo DNA methylation in Arabidopsis | https://www.ncbi.nlm.nih.gov/sra/?term=SRP059814 | Publicly available at NCBI Sequence Read Archive (accession no: SRP059814) |

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
