## [Decision Letter]

Thank you for submitting your article "Long Non-coding RNA Produced by RNA Polymerase V Determines Boundaries of Heterochromatin" for consideration by *eLife*. Your article has been favorably evaluated by Detlef Weigel (Senior Editor) and three reviewers, one of whom is a member of our Board of Reviewing Editors. The following individual involved in review of your submission has agreed to reveal her identity: Rebecca Mosher (Reviewer #3).

The reviewers have discussed the reviews with one another and the Reviewing Editor has drafted this decision to help you prepare a revised submission.

Summary:

Using RNA Immunoprecipitation coupled to high-throughput sequencing (RIP-seq), you have globally identified transcripts of RNA polymerase V (Pol V). You have further analyzed the correlations between Pol V transcription and Pol V binding, DNA methylation, siRNA accumulation and genomic features. Your analyses clarify several key questions regarding Pol V transcription and RNA-directed DNA methylation. You propose that Pol V transcription plays an important role in the determination of heterochromatin boundaries.

The reviewers agreed that such findings should in principle be published at the highest level. While the reviewers furthermore agreed that the data presented are clear, they also agreed that some conclusions went considerably beyond what the data showed directly. To bring the interpretation more in line with the data, they asked you to tone down your claims, and also to perform additional analyses to support your claims.

Essential revisions:

1) Analyze whether gene body methylation can direct Pol V transcription. This analysis can be easily done using the already available datasets and will address the question whether methylation alone is sufficient to direct Pol V.

2) The conclusion that AGO4 interacts with RNA exiting Pol V is not supported by sufficient data. Tone down this claim by rewording as reviewers #1 and #2 suggested or provide new data to strengthen it.

3) Examine the role of Pol IV in targeting RdDM to TE edges by performing analyses as done for Pol V. Alternatively, tone down the claim that Pol V is the main determinant for RdDM targeting and do not exclude the possibility that Pol IV also plays a similar role.

We have included the full reviews, which mention separate points for essential revisions. Reviewer #1 suggested web-lab experiments to validate some of your genome-wide analyses, but after discussion, we agree that these experiments are unnecessary. Meanwhile, we agree that other suggestions raised by reviewers #1 and #2 are valuable and most of them can be addressed by rewording. We ask you to take them into account as well.

*Reviewer #1:*

In the RNA-directed DNA methylation pathway, Pol V transcripts have been proposed as scaffold RNAs that recruit AGO4/siRNA effector complexes through base-pairing between the transcripts and siRNAs. In this manuscript, the authors present a comprehensive study that provides new evidence supporting previously proposed model for the role of Pol V transcripts at the effector stage of RdDM. Some of the findings advance the current understanding of RdDM and should be of general interest as well. My major criticism is that many statements made by the authors are too bold and not supported by sufficient data. The author should be more careful with interpretation of their seq data analyses.

1) The first two sections of Results can be organized in a more concise way. The authors should note the difference between "Pol V-associated RNA" and "Pol V transcript". For an example, in the first section, the authors mentioned that a considerable number of Pol V-associated RNAs were not Pol V-dependent, but in the second section, they concluded: "Pol V-associated RNAs are produced by Pol V".

2) The authors have not successfully identified the promoters of Pol V, thus the use of "The Pol V promoter" as the subtitle of the third section of Results is inappropriate.

3) The authors analyzed Pol V transcripts in *ago4* mutant and found that 1) the levels of Pol V transcripts were reduced in *ago4* and 2) average length of Pol V transcripts in *ago4* was similar to that in Col-0. As the authors mentioned, the reduction in Pol V transcript level in *ago4* could be from an indirect effect of *ago4* mutation on Pol V transcription. Thus, the authors should not conclude that AGO4 and IDN2 stabilize Pol V transcripts and use it as the subtitle of the section. The authors also concluded that Pol V transcripts are not sliced by AGO4, solely based on the data showing that average length of Pol V transcripts in *ago4* was similar to that in Col-0. The authors should consider an alternative interpretation of the data: the cleavage products of AGO4/siRNA were small and not detected by Pol V RIP-seq in Col-0 as there are many siRNAs that cover the entire region of a Pol transcript.

4) The seq data described in the section "siRNAs base pair with RNA exiting Pol V" basically supported a correlation between AGO4 binding to Pol V transcripts and RdDM but cannot be used to exclude the possibility that AGO4 binds to DNA. The authors provided an explanation for how AGO4 is prevented from binding to heterochromatin but did not explain how AGO4 is guided to regions outside heterochromatin. It is a bit strange that AGO4 was not ChIPed to RdDM regions. If AGO4 binds to Pol V transcripts, one would also expect AGO4 to be ChIPed onto Pol V transcribed regions through its binding to Pol V transcripts. In my opinion, without more biochemical data, it is premature to conclude that AGO4- associated siRNAs base pair with RNA exiting Pol V.

5) The authors found enrichment of Pol V transcription on the edges of longer transposons and concluded that Pol V is the main determinant for RdDM targeting the edges of transposons. As matter of fact, as the authors noted, Pol IV-dependent siRNAs were also enriched on the edges. Thus, Pol IV could also be a determinant.

6) In the third section of Results, the authors indicated that Pol V transcribed both strands and the transcription levels on both strands were correlated. However, in the last section of Results, strand preferences were detected for 5'-ends and 3'-ends of transposons. This needs to be clarified.

7) Most of the conclusions were drawn from genome-wide data analyses. The authors should use alternative approaches to validate at least some of the key points. For instance, the lengths of transcripts generated by internal Pol V transcription at some representative loci should be examined by RT-PCR analysis. Strand preference of Pol V transcription at selected loci should be confirmed by qRT-PCR. AGO4 binding to Pol V transcripts or DNA should be validated by RIP-RTPCR or ChIP-qPCR.

*Reviewer #2:*

In the manuscript entitled "Long Non-coding RNA Produced by RNA Polymerase V Determines the Boundaries of Heterochromatin" submitted to *eLife*, Böhmdorfer et al. perform RIP-seq of Pol V interacting RNAs in wild-type and pol v mutants. This dataset thus defines the transcripts that Pol V produces. The authors perform several analyses connecting their dataset to ChIP, RNA-seq and sRNA-seq datasets either already published or that they have produced themselves. This analysis shows that their dataset of Pol V transcripts is higher-resolution compared to Pol V ChIP previously published. The authors also use their dataset and informatic prowess to address several key questions in the RdDM field. Overall, this manuscript is very carefully written. I have split my comments into major concerns and analyses to be performed.

Major concerns:

1) The production and analysis of the Pol V-RIP dataset represents a technical accomplishment. However, my major concern for this manuscript is what has been biologically learned that we did not already know? This higher resolution data has improved our detection, but what was learned about RdDM? I think it will be important for the authors to highlight the strong conclusions from their dataset in the Abstract, and remove the Abstract parts that focus on the production of the dataset.

2) I found certain sections of the Methods lacking in critical detail. Were mock-IPs of any sort performed? Was mRNA or rRNA depleted in any way before library production? Most importantly, how were multi-mapping reads handled? Much of Pol V RNA comes from repetitive DNA, and how this is informatically handled will have large consequences on the observed data. This is particularly important for the data in Figure 7.

3) One of the strongest conclusions that could be made is that there is absolutely no evidence from this dataset that Pol II can substitute for Pol V in a pol v mutant. I think the authors should make this strong agreement to improve the conclusions they are drawing in this manuscript compared to their Pol V ChIP publication.

Analyses to be performed:

1) Subsection “The Pol V Promoter”: “Pol V transcripts were enriched on transposable elements in gene-rich environments”. I do not see evidence of this in Figure 2. First, the r2 value is very low. Second, the analysis performed does not look at gene-rich or poor environments, but rather at mRNA production (mRNA/TE), which is not the same thing. This analysis should be repeated in a more direct manner.

2) The evidence in Figure 2 makes me think that all regions regulated by MET1 are transcribed by Pol V. Does this include CG-context methylated genes? I think the investigation of these genes is critical to understanding Pol V. If they are transcribed by Pol V, then methylation alone, and not histone modification / heterochromatin is responsible for directing Pol V. If CG methylated genes are not transcribed, then methylation alone is not sufficient to direct Pol V. Either way, these genes should be explicitly examined as they are methylated regions that should not be targeted for RdDM nor formed into heterochromatin.

3) Subsection “AGO4 Binds Most Pol V Transcripts”: “…suggesting that AGO4 associates with most if not all Pol V transcripts.” I am not convinced of this claim by the data presented. Could the authors use a Venn diagram to determine the overlap and support this claim?

4) I do not like the display of Figure 4. If the authors wish to compare the RIP data in the *ago4* mutant, I suggest creating a scatter plot of each of the Pol V transcripts they annotated as dots, and then plot their RPM RIP values for wt on the X-axis and *ago4* on the Y axis. This will display the dataset in a more useful way than the boxplots or metaplot. See the following reference for an example: Panoramix enforces piRNA-dependent cotranscriptional silencing.

5) In the subsection “siRNAs Base Pair with RNA Exiting Pol V” the authors investigate strand bias. This is a worthwhile analysis, but it only informs the siRNA base pairing with RNA due to the models proposed, and does not directly test the "RNA exiting Pol V" per se. I would write this section strongly from the strand-bias point of view, but then only suggest that this supports an RNA-RNA interaction at a distance from the polymerase.

6) In the section on edges of heterochromatin, it seems to me that the combined activity of both Pol IV and Pol V function to create this edge enrichment. I would be interested to see if the analysis performed in Figure 7 repeated with data from Pol IV would show the same thing of transcription in from transposable element ends. I think this is particularly important, as this is a major conclusion that could be newly drawn from this dataset, and the title reflects this discovery as well.

*Reviewer #3:*

In this manuscript, Böhmdorfer and colleagues address the important question of Pol V activity by describing transcripts produced by RNA Pol V in *Arabidopsis*. They identify these transcripts through Pol V RIP-seq and further assess them in various ways, including assessment via qRT-PCR in *nrpe1* mutants or RIP-seq in *ago4* and *idn2* mutants, comparison of sense/antisense transcripts, and correlations with Pol V ChIP, DNA methylation, siRNA accumulation, and genomic features (TEs, histone modifications). The authors make a number of conclusions regarding Pol V activity and the larger mechanisms of RdDM. While some of their conclusions are based on indirect evidence, they are always careful to state alternative hypotheses. The data in this manuscript are of the highest quality and will undoubtedly be highly influential. The authors should be commended on a really beautiful piece of science – meticulous experimentation, thoughtful analysis, careful interpretation, clear writing, and beautiful figures.

---

## [Author Response]

*[…] The reviewers agreed that such findings should in principle be published at the highest level. While the reviewers furthermore agreed that the data presented are clear, they also agreed that some conclusions went considerably beyond what the data showed directly. To bring the interpretation more in line with the data, they asked you to tone down your claims, and also to perform additional analyses to support your claims.*

Essential revisions:

*1) Analyze whether gene body methylation can direct Pol V transcription. This analysis can be easily done using the already available datasets and will address the question whether methylation alone is sufficient to direct Pol V.*

We performed the requested analysis and show that CG-methylated genes are not transcribed by Pol V. This provides the important new conclusion that CG is required but not sufficient for Pol V transcription. This new result is shown in Figure 2 and explained in a new paragraph in the Results.

*2) The conclusion that AGO4 interacts with RNA exiting Pol V is not supported by sufficient data. Tone down this claim by rewording as reviewers #1 and #2 suggested or provide new data to strengthen it.*

We have extensively rewritten and toned down the relevant subsection of the Results. It is now split into two independent subsections, one focused on strand bias and the other on AGO4 binding to Pol V transcripts. The new wording does not use the presented results as a way to test if AGO4 interacts with RNA exiting Pol V. Instead, we test smaller hypotheses and only finish the second subsection with the following statement: “(…) our data are consistent with a speculative model where downstream components of RdDM interact with Pol V transcripts in close proximity to the Pol V complex and siRNAs base pair with RNA exiting the Pol V complex”. We left a paragraph in the Discussion explaining how our results support this model but made it clear that we only provide additional indirect evidence.

*3) Examine the role of Pol IV in targeting RdDM to TE edges by performing analyses as done for Pol V. Alternatively, tone down the claim that Pol V is the main determinant for RdDM targeting and do not exclude the possibility that Pol IV also plays a similar role.*

We performed the requested analysis using published datasets of Pol IV transcription and this new analysis is now described in the Results and shown in Figure 7—figure supplement 1. Briefly, we found that Pol IV is also enriched on the edges of TEs. However, we found no evidence of strand specificity. These new results are explained in the Results and mentioned in the Discussion. For clarity we removed results shown in Figure 7—figure supplement 1 panels C-E in the original manuscript.

*We have included the full reviews, which mention separate points for essential revisions. Reviewer #1 suggested web-lab experiments to validate some of your genome-wide analyses, but after discussion, we agree that these experiments are unnecessary. Meanwhile, we agree that other suggestions raised by reviewers #1 and #2 are valuable and most of them can be addressed by rewording. We ask you to take them into account as well.*

*Reviewer #1:*

*In the RNA-directed DNA methylation pathway, Pol V transcripts have been proposed as scaffold RNAs that recruit AGO4/siRNA effector complexes through base-pairing between the transcripts and siRNAs. In this manuscript, the authors present a comprehensive study that provides new evidence supporting previously proposed model for the role of Pol V transcripts at the effector stage of RdDM. Some of the findings advance the current understanding of RdDM and should be of general interest as well. My major criticism is that many statements made by the authors are too bold and not supported by sufficient data. The author should be more careful with interpretation of their seq data analyses.*

*1) The first two sections of Results can be organized in a more concise way. The authors should note the difference between "Pol V-associated RNA" and "Pol V transcript". For an example, in the first section, the authors mentioned that a considerable number of Pol V-associated RNAs were not Pol V-dependent, but in the second section, they concluded: "Pol V-associated RNAs are produced by Pol V".*

In the revised manuscript the term “Pol V transcripts” is only introduced after Pol V-associated transcripts are shown to depend on Pol V and to be produced from Pol V-bound loci. We clarified that transcripts, which are not dependent on Pol V “(…) are unlikely to be associated with Pol V”.

We considered ways to make the first two sections more concise. However, we thought that, even though specialists could consider some of the information provided in those sections as self-explanatory, the broad audience of *eLife* could benefit from a more detailed explanation of our experimental approaches that create the foundation for the rest of the manuscript.

*2) The authors have not successfully identified the promoters of Pol V, thus the use of "The Pol V promoter" as the subtitle of the third section of Results is inappropriate.*

We changed the subtitle to “Pol V Regulatory Elements”.

*3) The authors analyzed Pol V transcripts in ago4 mutant and found that 1) the levels of Pol V transcripts were reduced in ago4 and 2) average length of Pol V transcripts in ago4 was similar to that in Col-0. As the authors mentioned, the reduction in Pol V transcript level in ago4 could be from an indirect effect of ago4 mutation on Pol V transcription. Thus, the authors should not conclude that AGO4 and IDN2 stabilize Pol V transcripts and use it as the subtitle of the section. The authors also concluded that Pol V transcripts are not sliced by AGO4, solely based on the data showing that average length of Pol V transcripts in ago4 was similar to that in Col-0. The authors should consider an alternative interpretation of the data: the cleavage products of AGO4/siRNA were small and not detected by Pol V RIP-seq in Col-0 as there are many siRNAs that cover the entire region of a Pol transcript.*

We changed the title of the sub-section to “AGO4 and IDN2 Enhance the Accumulation of Pol V Transcripts”. We also added the suggested alternative interpretation by stating that “(…) AGO4 slicing products originating from Pol V transcripts could be undetectable in our assay due to their size or loss of association with the Pol V complex”.

Additional modifications to this subsection have been made in response to Reviewer #2 and are described below.

*4) The seq data described in the section "siRNAs base pair with RNA exiting Pol V" basically supported a correlation between AGO4 binding to Pol V transcripts and RdDM but cannot be used to exclude the possibility that AGO4 binds to DNA. The authors provided an explanation for how AGO4 is prevented from binding to heterochromatin but did not explain how AGO4 is guided to regions outside heterochromatin. It is a bit strange that AGO4 was not ChIPed to RdDM regions. If AGO4 binds to Pol V transcripts, one would also expect AGO4 to be ChIPed onto Pol V transcribed regions through its binding to Pol V transcripts. In my opinion, without more biochemical data, it is premature to conclude that AGO4- associated siRNAs base pair with RNA exiting Pol V.*

As explained above in our response to Essential Revisions, this paragraph has been rewritten and toned down. We now do not make this conclusion, instead we mention in the Results that our data provide additional support for this speculative model and further expand on those speculations in the Discussion. We also directly state that AGO4/siRNA-DNA interaction remains possible.

It is important to point out that AGO4 does in fact generally ChIP to RdDM targets, which we published before and further confirm on Figure 3. The exclusion observed in Figure 6 is a local phenomenon, which we only observed on the edges of domains showing high levels of H3K9me2.

*5) The authors found enrichment of Pol V transcription on the edges of longer transposons and concluded that Pol V is the main determinant for RdDM targeting the edges of transposons. As matter of fact, as the authors noted, Pol IV-dependent siRNAs were also enriched on the edges. Thus, Pol IV could also be a determinant.*

As explained in our response to Essential Revisions, we tested this possibility more directly by analyzing the presence of Pol IV transcripts on the edges of transposons and now show that both Pol IV and Pol V are involved in this process. These new data are shown in Figure 7—figure supplement 1.

*6) In the third section of Results, the authors indicated that Pol V transcribed both strands and the transcription levels on both strands were correlated. However, in the last section of Results, strand preferences were detected for 5'-ends and 3'-ends of transposons. This needs to be clarified.*

We clarified this apparent inconsistency by adding the following sentence in the last subsection of Results: “Although Pol V tends to transcribe both strands of DNA (Figure 2), a subset of Pol V transcripts is enriched on one strand (Figure 6—figure supplement 1) indicating that limited strand preference of Pol V may be involved in determining boundaries of heterochromatin.”

*7) Most of the conclusions were drawn from genome-wide data analyses. The authors should use alternative approaches to validate at least some of the key points. For instance, the lengths of transcripts generated by internal Pol V transcription at some representative loci should be examined by RT-PCR analysis. Strand preference of Pol V transcription at selected loci should be confirmed by qRT-PCR. AGO4 binding to Pol V transcripts or DNA should be validated by RIP-RTPCR or ChIP-qPCR.*

We followed the editorial decision letter and did not add these locus-specific validations. We should explain that we invested a substantial amount of time and resources into directly identifying ends of Pol V transcripts. Unfortunately, presence of Pol II transcripts introduced some level of ambiguity and we decided not to publish these results.

*Reviewer #2:*

*In the manuscript entitled "Long Non-coding RNA Produced by RNA Polymerase V Determines the Boundaries of Heterochromatin" submitted to eLife, Böhmdorfer et al. perform RIP-seq of Pol V interacting RNAs in wild-type and pol v mutants. This dataset thus defines the transcripts that Pol V produces. The authors perform several analyses connecting their dataset to ChIP, RNA-seq and sRNA-seq datasets either already published or that they have produced themselves. This analysis shows that their dataset of Pol V transcripts is higher-resolution compared to Pol V ChIP previously published. The authors also use their dataset and informatic prowess to address several key questions in the RdDM field. Overall, this manuscript is very carefully written. I have split my comments into major concerns and analyses to be performed.*

*Major concerns:*

*1) The production and analysis of the Pol V-RIP dataset represents a technical accomplishment. However, my major concern for this manuscript is what has been biologically learned that we did not already know? This higher resolution data has improved our detection, but what was learned about RdDM? I think it will be important for the authors to highlight the strong conclusions from their dataset in the Abstract, and remove the Abstract parts that focus on the production of the dataset.*

We have rewritten the Abstract to stress most novel aspects of our work.

*2) I found certain sections of the Methods lacking in critical detail. Were mock-IPs of any sort performed? Was mRNA or rRNA depleted in any way before library production? Most importantly, how were multi-mapping reads handled? Much of Pol V RNA comes from repetitive DNA, and how this is informatically handled will have large consequences on the observed data. This is particularly important for the data in Figure 7.*

We expanded the Methods section, which now provides detailed information about the IP protocol and mapping. We also combined all methods in the main manuscript text. Briefly, mock IPs were tested by locus-specific assays but not used for sequencing, there was no rRNA/mRNA depletion and reads mapping to more than one position in the genome were mapped only once.

*3) One of the strongest conclusions that could be made is that there is absolutely no evidence from this dataset that Pol II can substitute for Pol V in a pol v mutant. I think the authors should make this strong agreement to improve the conclusions they are drawing in this manuscript compared to their Pol V ChIP publication.*

We added this important conclusion in a new paragraph in the Discussion section.

*Analyses to be performed:*

*1) Subsection “The Pol V Promoter”: “Pol V transcripts were enriched on transposable elements in gene-rich environments”. I do not see evidence of this in Figure 2. First, the r2 value is very low. Second, the analysis performed does not look at gene-rich or poor environments, but rather at mRNA production (mRNA/TE), which is not the same thing. This analysis should be repeated in a more direct manner.*

This analysis is indeed to some extent confusing. Because it is mostly confirmatory, we decided to remove this analysis from the revised manuscript.

*2) The evidence in Figure 2 makes me think that all regions regulated by MET1 are transcribed by Pol V. Does this include CG-context methylated genes? I think the investigation of these genes is critical to understanding Pol V. If they are transcribed by Pol V, then methylation alone, and not histone modification / heterochromatin is responsible for directing Pol V. If CG methylated genes are not transcribed, then methylation alone is not sufficient to direct Pol V. Either way, these genes should be explicitly examined as they are methylated regions that should not be targeted for RdDM nor formed into heterochromatin.*

As explained in our response to Essential Revisions, we performed this analysis and show that CG-methylated genes are not transcribed by Pol V and, therefore, CG methylation is not sufficient for Pol V transcription (Figure 2). At the same time, we decided to remove a former Figure 2, which served mostly a confirmatory role.

*3) Subsection “AGO4 Binds Most Pol V Transcripts”: “…suggesting that AGO4 associates with most if not all Pol V transcripts.” I am not convinced of this claim by the data presented. Could the authors use a Venn diagram to determine the overlap and support this claim?*

This conclusion is primarily based on the heatmap shown in Figure 3, which was not properly explained in the original manuscript. We have rewritten this part of the results to explain our reasoning. We do not believe that a Venn diagram would provide a more useful insight because it relies on an arbitrary criterion of the presence or absence of AGO4 RIP-seq. The heatmap we show leads to the same conclusion, but also shows the variability of the AGO4 RIP-seq signal.

*4) I do not like the display of Figure 4. If the authors wish to compare the RIP data in the ago4 mutant, I suggest creating a scatter plot of each of the Pol V transcripts they annotated as dots, and then plot their RPM RIP values for wt on the X-axis and ago4 on the Y axis. This will display the dataset in a more useful way than the boxplots or metaplot. See the following reference for an example: Panoramix enforces piRNA-dependent cotranscriptional silencing.*

We generated the suggested scatterplots, which show the same trends as seen on the boxplots in Figure 4. We are however not convinced that this data presentation is more useful than boxplots or metaplots and would like to keep Figure 4AB unchanged and show the new scatterplots in Figure 4—figure supplement 1.

*5) In the subsection “siRNAs Base Pair with RNA Exiting Pol V” the authors investigate strand bias. This is a worthwhile analysis, but it only informs the siRNA base pairing with RNA due to the models proposed, and does not directly test the "RNA exiting Pol V" per se. I would write this section strongly from the strand-bias point of view, but then only suggest that this supports an RNA-RNA interaction at a distance from the polymerase.*

As explained above in our responses to Essential Revisions and Reviewer #1, this paragraph has been rewritten and toned down. We split the original sub-section into two shorter sub-sections, one of which is focused on strand bias without any mention of siRNA-lncRNA base-pairing. We now only mention in the Results that our data provide additional support for the speculative model and further expand on those speculations in the Discussion.

*6) In the section on edges of heterochromatin, it seems to me that the combined activity of both Pol IV and Pol V function to create this edge enrichment. I would be interested to see if the analysis performed in Figure 7 repeated with data from Pol IV would show the same thing of transcription in from transposable element ends. I think this is particularly important, as this is a major conclusion that could be newly drawn from this dataset, and the title reflects this discovery as well.*

As explained in our responses to Essential Revisions and Reviewer #1, we performed the analysis of Pol IV transcripts and found that indeed both Pol IV and Pol V are preferentially present on the edges of transposons. Interestingly, only Pol V shows directionality, which we explain as an effect of RDR2 association with Pol IV. These new results are shown in Figure 7—figure supplement 1 and explained in the Results section.